# Towards Provably Efficient Learning of Extensive-Form Games with Imperfect Information and Linear Function Approximation

## Abstract

We study two-player zero-sum imperfect information extensive-form games (IIEFGs) with linear functional approximation. In particular, we consider linear IIEFGs in the formulation of partially observable Markov games (POMGs) with unknown transition and bandit feedback, in which the reward function admits a linear structure. To tackle the partial observation of this problem, we propose a linear loss estimator based on the *composite* features of information set-action pairs. Through integrating this loss estimator with the online mirror descent (OMD) framework and delicate analysis of the stability term in the linear case, we prove the $\widetilde{\mathcal{O}}(\sqrt{(d + 1/\rho)HX^2T})$ regret upper bound of our algorithm, where $H$ is the horizon length, $X$ is the cardinality of the information set space, $d$ is the ambient dimension of the feature mapping, and $\rho$ is the minimum eigenvalue of the feature covariance matrix generated by the exploration policy. Additionally, by leveraging the "transitions" over information set-actions, we propose another algorithm based on the follow-the-regularized-leader (FTRL) framework, attaining a regret bound of $\widetilde{\mathcal{O}}(\sqrt{H^2 d\lambda T})$, where $\lambda$ is a quantity depends on the game tree structure. Moreover, we prove that our FTRL-based algorithm also achieves the $\widetilde{\mathcal{O}}(\sqrt{HXdT})$ regret with a different initialization of parameters. Further, we provide an $\Omega(\sqrt{d \min(d, H)T})$ regret lower bound for this problem. To the best of our knowledge, we present the first line of algorithms studying learning IIEFGs with linear function approximation.

## 1 Introduction

In imperfect information games (IIGs), players are limited in their knowledge of the true state of play when making moves. This opacity allows for intricate strategic maneuvers like bluffing, as players can hide private information from opponents. In particular, the notion of imperfect-information extensive-form games (IIEFGs) (Kuhn, 1953) simultaneously enables imperfect information and sequential play, which thus characterizes a large amount of modeling real-world imperfect information games including Poker (Heinrich et al., 2015; Moravčík et al., 2017; Brown & Sandholm, 2018), Bridge (Tian et al., 2020), Scotland Yard (Schmid et al., 2021) and Mahjong (Li et al., 2020; Kurita & Hoki, 2021; Fu et al., 2022). There has been an extensive line of works on regret minimization or finding the Nash equilibrium (NE) (Nash Jr, 1950) of IIEFGs. Under perfect recall condition, when the full knowledge of the game is known, existing works solve this problem by linear programming (Koller & Megiddo, 1992; Von Stengel, 1996; Koller et al., 1996), first-order optimization methods (Hoda et al., 2010a; Kroer et al., 2015a; 2018; Munos et al., 2020; Lee et al., 2021; Liu et al., 2022), and counterfactual regret minimization (Zinkevich et al., 2007; Lanctot et al., 2009; Johanson et al., 2012; Tammelin, 2014; Schmid et al., 2019; Burch et al., 2019; Liu et al., 2022).

When the full knowledge of the game is not known a priori or only partial knowledge of the game is revealed, the problem will be much more challenging and is typically tackled through learning from the random observations accrued during repeated plays of the game. In this line of works, two-player zero-sum IIEFGs have been addressed via equipping online mirror descent (OMD) or follow-the-regularized-leader (FTRL) frameworks with loss estimations (Farina et al., 2021; Kozuno et al., 2021; Bai et al., 2022; Fiegel et al., 2023) and Monte-Carlo counterfactual regret minimization (Lanctot et al., 2009; Farina et al., 2020; Farina & Sandholm, 2021). Amongst these work, Bai

et al. (2022) leverage OMD with "balanced exploration policies" to achieve the first $\widetilde{\mathcal{O}}(\sqrt{H^3 X A T})$ regret bound, where $H$ is the horizon length, $X$ is the cardinality of the information set space, $A$ is the cardinality of the action space and $T$ is the number of episodes. Notably, this sample complexity matches the information-theoretic lower bound on all factors but $H$ up to logarithmic factors. Subsequently, Fiegel et al. (2023) further improve the bound to $\widetilde{\mathcal{O}}(\sqrt{X A T})$, with optimal dependency on all factors up to logarithmic factors, using FTRL with "balanced transitions".

Though significant advances have emerged in learning two-player zero-sum IIEFGs, the existing sample complexities of all works depend on $X$ and $A$. In practice, however, $X$ and/or $A$ might be very large, particularly in large-scale IIEFGs, which makes the above sample complexities vacuous. This issue, which is typically called the *curse of dimensionality*, has also emerged in various problems beyond IIEFGs. To cope with this issue, a common approach is *function approximation*, which approximates the observations on experienced information sets and/or actions with sharing parameters and generalizes them onto unseen information sets and/or actions. Indeed, for practitioners in the area of IIEFGs (*e.g.*, (Moravčík et al., 2017; Brown et al., 2019)), function approximation using, for example, deep neural networks, has made significant progress in solving large-scale IIEFGs. Yet, the theoretical guarantees of learning algorithms with function approximation for IIEFGs still remain open and we are still far from understanding them well. On the other hand, in the more amenable sequential-decision making problems including (single-agent) reinforcement learning (RL) (Ayoub et al., 2020; Jin et al., 2020; Zhou et al., 2021; He et al., 2023) and Markov games (MGs) with perfect information (Chen et al., 2022; Ni et al., 2023; Wang et al., 2023; Cui et al., 2023), significant advances have emerged in understanding the theoretical guarantees of algorithms with function approximation. Therefore, the above two facts naturally motivate us to ask the following question:

*Does there exist a provably efficient algorithm for IIEFGs in the function approximation setting?*

In this paper, we give an affirmative answer to the above question for IIEFGs with linear function approximation, in the *offline* setting[1]. In specific, we consider IIEFGs in the formulation of partially observable Markov games (POMGs) with unknown transition and unknown rewards while admitting a linear structure over the reward functions. This problem is challenging in that both players are unaware of the current underlying state since only the current information set rather than the state is observable, which poses substantial difficulties in exploiting the linear structure of the reward functions, as the current feature corresponding to the current state is unknown. To address this problem and also establish efficient algorithms for learning IIEFGs with linear function approximation, in this paper, we make the following contributions:

- To learn the unknown parameter that linearly parameterizes the reward functions, we instead utilize a kind of *composite* reward features, weighted by the transitions and opponent's policy. Intuitively, composite reward features can be seen as features of corresponding information set-actions. Equipped with the composite reward features, we further propose the first least-squares loss estimator for this problem and prove its unbiasedness (see Section 3.1 for details).

- Based on the least-squares loss estimator, we then propose the least-squares online mirror descent (LSOMD) algorithm that attains the $\widetilde{\mathcal{O}}(\sqrt{(d + 1/\rho) H X^2 T})$ regret bound, where $d$ is the ambient dimension of the feature mapping and $\rho := \min_{t \in [T], h \in [H]} \lambda_{\min}(\boldsymbol{Q}_{\pi^t, h})$ with $\boldsymbol{Q}_{\pi^t, h}$ being as the feature covariance matrix induced by exploration policy $\pi^t$ at step $h$. Compared to the computation and regret analysis of OMD in tabular IIEFGs (Kozuno et al., 2021; Bai et al., 2022; Fiegel et al., 2023) that heavily depends on the sparsity of the importance-weighted loss estimate, however, our case intrinsically requires new ingredients to solve both aspects, due to the leverage of the linear structure. The key insight is to solve the computation and also bound the stability term of LSOMD by the log-partition function $\log Z_1^t$, which is in turn bounded by the expectation of the element-wise product of all the random vectors sampled from all the categorical distributions along paths from the root node (see Section 3.3 for details).

- Via integrating our proposed linear loss estimator, the solution to the optimization problem based on the log-partition function $\log Z_1^t$ and the idea of "balanced transition", which shares a similar spirit as Bai et al. (2022); Fiegel et al. (2023), we additionally propose the least-squares follow-the-regularized-leader (LSFTRL) algorithm. Let $p_{1:h}^{\nu}(x_h) =$

---

[1]By "offline" we refer to that the feature vectors of state-action weighted by min-player's policy $\nu^t$ in episode $t$ (as well as transitions) are accessible to the max-player before the $t$-th episode starts. Please see Section 2 for more discussions.

Table 1: Comparisons of regret bounds with most related works studying IIEFGs when the full knowledge of the game is not known a priori.

| Algorithm | Setting | Regret |
|---|---|---|
| IXOMD (Kozuno et al., 2021) | Online | $\widetilde{\mathcal{O}}(HX\sqrt{AT})$ |
| Balanced OMD/CFR (Bai et al., 2022) | | $\widetilde{\mathcal{O}}(\sqrt{H^3XAT})$ |
| Balanced FTRL (Fiegel et al., 2023) | | $\widetilde{\mathcal{O}}(\sqrt{XAT})$ |
| LSOMD (this paper) | Offline[1] | $\widetilde{\mathcal{O}}(\sqrt{(d+1/\rho)HX^2T})$ [2] |
| LSFTRL (this paper) | | $\widetilde{\mathcal{O}}(\sqrt{H^2d\lambda T})/\widetilde{\mathcal{O}}(\sqrt{HXdT})$ [3] |
| Lower bound (this paper) | - | $\Omega(\sqrt{d\min(d,H)T})$ |

[1] See Section 2 for the definition of our *offline* setting.
[2] See Assumption 3.2 for the definition of $\rho$.
[3] The $\lambda$ in the former bound depends on the game tree structure, defined in Assumption 4.1. The latter bound is obtained by the same algorithm but with a different initiation of parameters.

$\sum_{s_h \in x_h} p_{1:h}(s_h)\nu_{1:h-1}(y(s_{h-1}), b_{h-1})$ be the sequence-form representation of the "transition" over $\mathcal{X}_h \times \mathcal{A} \times \mathcal{X}_{h+1}$ induced by the environment transition $\mathbb{P} = \{p_h\}_{h=0}^{H-1}$ and opponent's policy $\nu$. Under the assumption that for any $t \in [T]$, $h \in [H]$ and $x_1, x_2 \in \mathcal{X}_h$, $p_{1:h}^{\nu^t}(x_1)/p_{1:h}^{\star}(x_2) \leq \lambda$ with $p_{1:h}^{\star}$ being the sequence-form representation of the chosen "balanced transition" over $\mathcal{X}_h \times \mathcal{A} \times \mathcal{X}_{h+1}$ in LSFTRL, we prove that the regret upper bound of LSFTRL is of order $\widetilde{\mathcal{O}}(\sqrt{H^2d\lambda T})$. With a different initialization of "balanced transition" parameter that is not leveraged in previous works and a refined analysis on the stability term, we also prove that LSFTRL enjoys a $\widetilde{\mathcal{O}}(\sqrt{HXdT})$ regret (see Section 4.2 for details).

## 1.1 RELATED WORK

**Partially observable Markov games (POMGs)** With perfect information, learning MGs dates back to the work of Littman & Szepesvári (1996) and has been well-studied (Littman, 2001; Greenwald & Hall, 2003; Hu & Wellman, 2003; Hansen et al., 2013; Sidford et al., 2018; Lagoudakis & Parr, 2002; Pérolat et al., 2015; Fan et al., 2020; Jia et al., 2019; Cui & Yang, 2021; Zhang et al., 2021; Bai & Jin, 2020; Liu et al., 2021; Zhou et al., 2021; Song et al., 2022; Li et al., 2022; Xiong et al., 2022; Wang et al., 2023; Cui et al., 2023). When only with imperfect information but the full model of the game (*i.e.*, transitions and rewards) is known, existing works can be categorized into three lines. The first line uses sequence-form policies to reformulate this problem as a linear program (Koller & Megiddo, 1992; Von Stengel, 1996; Koller et al., 1996). The second line considers solving the minimax optimization problem directly by first-order algorithms (Hoda et al., 2010a; Kroer et al., 2015a; 2018; Munos et al., 2020; Lee et al., 2021; Liu et al., 2022). The last line of works tackles this problem using counterfactual regret minimization (CFR), which minimizes counterfactual regrets locally at each information set (Zinkevich et al., 2007; Lanctot et al., 2009; Johanson et al., 2012; Tammelin, 2014; Schmid et al., 2019; Burch et al., 2019; Liu et al., 2022). When the model of the game is not known or only partial knowledge of the game is accessible, the Monte-Carlo CFR algorithm proposed by Lanctot et al. (2009) attains the first $\varepsilon$-NE result in this problem. Subsequently, this framework is further generalized by Farina et al. (2020); Farina & Sandholm (2021). Besides, the other line of works considers combining OMD and FTRL with importance-weighted loss estimator (Farina et al., 2021; Kozuno et al., 2021; Bai et al., 2022; Fiegel et al., 2023) to tackle this problem. Remarkably, Bai et al. (2022) obtain the $\widetilde{\mathcal{O}}(\sqrt{H^3XAT})$ regret by using "balanced" dilated KL as the distance metric. With an analogous notion of "balanced" transition, Fiegel et al. (2023) finally achieve the regret of order $\widetilde{\mathcal{O}}(\sqrt{XAT})$, matching the lower bound up to logarithmic factors.

**Markov games with Function Approximation** To cope with the issue of the curse of dimensionality in MGs, there has been growing research interest in learning MGs in the function approximation setting recently (Xie et al., 2020; Chen et al., 2022; Xiong et al., 2022; Jin et al., 2022; Wang et al., 2023; Cui et al., 2023; Ni et al., 2023; Zhang et al., 2023). In particular, Xie et al. (2020) assume

both the transition and the reward functions of the episodic two-player zero-sum MGs are linearly realizable and achieve an $\widetilde{\mathcal{O}}(\sqrt{d^3 H^4 T})$ regret. More recent works generally fall into two categories. The first category aims to relax the assumption of linear function approximation by studying MGs in general function approximation (Xiong et al., 2022; Jin et al., 2022; Ni et al., 2023) and the other category of works focuses on learning general-sum MGs (Wang et al., 2023; Cui et al., 2023; Ni et al., 2023; Zhang et al., 2023). However, we note that all these works study *perfect information* MGs with function approximation, and (to our knowledge) there are no existing works studying *partially observable* MGs with function approximation, which is the main focus of our work.

## 2 PRELIMINARIES

Following previous works (Kozuno et al., 2021; Bai et al., 2022), in this work, we also study IIEFGs in the formulation of POMGs, the preliminaries of which are introduced in this section.

**Partially Observable Markov Games** An episodic, finite-horizon, two-player, zero-sum POMG is denoted by $\text{POMG}(\mathcal{S}, \mathcal{X}, \mathcal{Y}, \mathcal{A}, \mathcal{B}, H, \mathbb{P}, r)$, in which

- $H$ is the length of the horizon;
- $\mathcal{S} = \bigcup_{h \in [H]} \mathcal{S}_h$ is a finite state space with cardinality $S = \sum_{h=1}^{H} S_h$ and $|\mathcal{S}_h| = S_h$;
- $\mathcal{X} = \bigcup_{h \in [H]} \mathcal{X}_h$ and $\mathcal{Y} = \bigcup_{h \in [H]} \mathcal{Y}_h$ are the spaces of information sets (short for *infosets* in the following paper) for the *max-player* and *min-player*, respectively. Specifically, the cardinality $X$ of $\mathcal{X}$ satisfies $X := \sum_{h=1}^{H} X_h$ with $|\mathcal{X}_h| = X_h$ and the cardinality $Y$ of $\mathcal{Y}$ satisfies $Y := \sum_{h=1}^{H} Y_h$ with $|\mathcal{Y}_h| = Y_h$;
- $\mathcal{A}$ with $|\mathcal{A}| = A$ and $\mathcal{B}$ with $|\mathcal{B}| = B$ are the finite action spaces for the max-player and min-player, respectively;
- $\mathbb{P} = \{p_0(\cdot) \in \Delta_{\mathcal{S}_1}\} \cup \{p_h(\cdot \mid s_h, a_h, b_h) \in \Delta_{\mathcal{S}_{h+1}}\}_{(s_h, a_h, b_h) \in \mathcal{S}_h \times \mathcal{A} \times \mathcal{B}, h \in [H-1]}$ are the transition probability functions[2], with $p_0(\cdot)$ being the probability distribution of the initial states, and $p_h(s_{h+1}|s_h, a_h, b_h)$ being the probability of transmitting to the next state $s_{h+1}$ conditioned on $(s_h, a_h, b_h)$ at step $h$;
- $r = \{r_h(s_h, a_h, b_h) \in [-1, 1]\}_{(s_h, a_h, b_h) \in \mathcal{S}_h \times \mathcal{A} \times \mathcal{B}}$ are the stochastic reward functions with $\bar{r}_h(s_h, a_h, b_h)$ as means.

**Learning Protocol** To begin with, we denote by $\mu := \{\mu_h\}_{h \in [H]}$ with $\mu_h^t : \mathcal{X}_h \to \Delta_{\mathcal{A}}$ the max-player's stochastic policy and by $\Pi_{\max}$ the set of the policies of the max-player. The min-player's stochastic policy $\nu$ and the set of the policies of the min-player $\Pi_{\min}$ are defined similarly. The game proceeds in $T$ episodes. At the beginning of episode $t$, the max-player chooses a stochastic policy $\mu_t \in \Pi_{\max}$. And similarly, the min-player chooses $\nu_t \in \Pi_{\min}$. Then, an initial state $s_1^t$ will be sampled from $p_0$. At each step $h$, the max-player, and min-player will observe their infoset $x_h^t := x(s_h^t)$ and $y_h^t := y(s_h^t)$ respectively, but *without* observing $s_h^t$. Conditioned on $x_h^t$, the max-player will sample and execute an action $a_h^t \sim \mu_h^t(\cdot|x_h)$. Simultaneously, the min-player will take action $b_h^t \sim \nu_h^t(\cdot|y_h)$. Subsequently, the game will transit to the next state $s_{h+1}^t$, which is drawn from $p_h(\cdot|s_h^t, a_h^t, b_h^t)$. Also, the max-player and min-player will receive rewards $r_h^t := r_h(s_h^t, a_h^t, b_h^t)$ and $-r_h^t$ respectively. The episode will terminate after taking actions $a_H^t$ and $b_H^t$ conditioned on $x_H^t$ and $y_H^t$ respectively, *i.e.*, the game will terminate in $H$ steps.

**Perfect Recall and Tree Structure** As in previous works (Kozuno et al., 2021; Bai et al., 2022; Fiegel et al., 2023), we also suppose that the POMGs satisfy the *tree structure* and the *perfect recall* condition (Kuhn, 1953). In specific, the tree structure indicates that for any $h = 2, \ldots, H$ and $s_h \in \mathcal{S}$, there exists a *unique* trajectory $(s_1, a_1, b_1, \ldots, s_{h-1}, a_{h-1}, b_{h-1})$ leading to $s_h$. Besides, perfect recall condition holds for each player if for any $h = 2, \ldots, H$ and any infoset $x_h \in \mathcal{X}_h$ of the max-player, there exists a *unique* history $(x_1, a_1, \ldots, x_{h-1}, a_{h-1})$ leading to $x_h$ and similarly for the min-player. In addition, we denote by $C_{h'}(x_h, a_h) \subset \mathcal{X}_{h'}$ the descendants of $(x_h, a_h)$ at step $h' \geq h$. With slightly abuse of notations, we also let $C_{h'}(x_h) := \cup_{a_h \in \mathcal{A}} C_{h'}(x_h, a_h)$ and $C(x_h, a_h) := C_{h+1}(x_h, a_h)$.

---

[2]While in some games, $\{p_h\}_{h=1}^{H-1}$ might be time-homogeneous, *i.e.*, $\{p_h\}_{h=1}^{H-1}$ does not depend on $h$, we retain the dependence on $h$ in our notations as it allows the results to be applicable more broadly without too much additional efforts in the analysis, following previous works (Bai et al., 2022; Fiegel et al., 2023).

**Sequence-form Representations** In addition, for any pair of product policy $(\mu, \nu)$, the tree structure and perfect recall condition enable the *sequence-form representations* of the reaching probability of state-action $(s_h, a_h, b_h)$:

$$\mathbb{P}^{\mu,\nu}(s_h, a_h, b_h) = p_{1:h}(s_h)\mu_{1:h}(x(s_h), a_h)\nu_{1:h}(y(s_h), b_h), \tag{1}$$

where $p_{1:h}(s_h)$ is the sequence-form transition probability defined as $p_{1:h}(s_h) = p_0(s_1)\prod_{h' \leq h-1} p_{h'}(s_{h'+1} \mid s_{h'}, a_{h'}, b_{h'})$, and $\mu_{1:h}(\cdot, \cdot)$ and $\nu_{1:h}(\cdot, \cdot)$ are the sequence-form policies satisfying $\mu_{1:h}(x_h, a_h) := \prod_{h'=1}^{h} \mu_{h'}(a_{h'} \mid x_{h'})$ and $\nu_{1:h}(y_h, b_h) := \prod_{h'=1}^{h} \nu_{h'}(b_{h'} \mid y_{h'})$. Therefore, we slightly abuse the meanings of $\mu$ and $\nu$ by viewing $\mu = \{\mu_{1:h}\}_{h\in[H]}$ and $\nu = \{\nu_{1:h}\}_{h\in[H]}$ as *realization plans* (Von Stengel, 1996). Under sequence-form representations, it is then clear that $\Pi_{\max}$ is a convex compact subspace of $\mathbb{R}^{XA}$ satisfying constraints $\mu_{1:h}(x_h, a_h) \geq 0$ and $\sum_{a_h \in \mathcal{A}} \mu_{1:h}(x_h, a_h) = \mu_{1:h-1}(x_{h-1}, a_{h-1})$ with $(x_{h-1}, a_{h-1})$ being such that $x_h \in C(x_{h-1}, a_{h-1})$ (understanding $\mu_{1:0}(x_0, a_0) = p(\emptyset) = 1$).

**POMGs with Linear Function Approximation** We now introduce the linear realizability assumption over the reward functions of POMGs, detailed as follows.

**Assumption 2.1** (Linear Rewards in POMGs). *The reward function $r$ in* $\text{POMG}(\mathcal{S}, \mathcal{X}, \mathcal{Y}, \mathcal{A}, \mathcal{B}, H, \mathbb{P}, r)$ *is linearly realizable with a known feature mapping* $\phi : \mathcal{S} \times \mathcal{A} \times \mathcal{B} \to \mathbb{R}^d$ *if for each $h \in [H]$, there exists an unknown parameter vector $\boldsymbol{\theta}_h \in \mathbb{R}^d$ such that for any $(s_h, a_h, b_h) \in \mathcal{S}_h \times \mathcal{A} \times \mathcal{B}$, it holds that $\bar{r}_h(s_h, a_h, b_h) = \langle \phi(s_h, a_h, b_h), \boldsymbol{\theta}_h \rangle$. In addition, we further assume that $\|\boldsymbol{\theta}_h\|_2 \leq \sqrt{d}$, $\sup_{(s_h, a_h, b_h) \in \mathcal{S}_h \times \mathcal{A} \times \mathcal{B}} \|\phi(s_h, a_h, b_h)\|_2 \leq 1$, and $\{\phi(s_h, a_h, b_h)\}_{(s_h, a_h, b_h) \in \mathcal{S}_h \times \mathcal{A} \times \mathcal{B}}$ spans $\mathbb{R}^d$, for any $h \in [H]$.*

Similar assumptions imposed over reward functions can also be seen in linear Markov games (Xie et al., 2020). But, again, as we shall see in Section 3.1, the imperfect information in POMGs brings significant difficulty in utilizing the linear structure over the reward functions compared with its fully observable counterpart. We also note that the regularity assumption imposed over $\phi(\cdot, \cdot, \cdot)$ and $\boldsymbol{\theta}_h$ is only for the purpose of normalization, and the assumption that $\mathbb{R}^d$ is spanned by the feature vectors is for convenience only (Lattimore & Szepesvári, 2020).

**Regret Minimization** For any product policy $(\mu, \nu)$, the value function of $(\mu, \nu)$ is defined as

$$V^{\mu,\nu} = \mathbb{E}\left[\sum_{h=1}^{H} r_h(s_h, a_h, b_h)\Big|\mu, \nu, \mathbb{P}\right], \tag{2}$$

where the expectation is taken over the randomness of the underlying state transitions and the policies of both players. In this paper, we consider the learning objective of regret minimization. Without loss of generality, we consider the case where the max-player is the learning agent, and the min-player is the (potentially adversarial) opponent, who might choose her policy $\nu^t$ arbitrarily, probably based on all the history information (including the knowledge of $\{\mu_k\}_{k=1}^{t-1}$) before episode $t$. In specific, the max-player aims to design policies $\{\mu^t\}_{t=1}^{T}$ to minimize the *pseudo-regret* (*regret* for short) compared with the best fixed policy $\mu^\dagger$ in hindsight, defined as

$$\mathfrak{R}_{\max}^T = \max_{\mu^\dagger \in \Pi_{\max}} \mathbb{E}\left[\sum_{t=1}^{T}\left(V^{\mu^\dagger,\nu^t} - V^{\mu^t,\nu^t}\right)\right]. \tag{3}$$

In this work, we consider the regret minimization for the max-player in the *offline* setting, in which the max-player has access to the feature vectors of state-action weighted by min-player's policy $\nu^t$ in episode $t$ (as well as transitions) before the $t$-th episode starts [3]. Note that this is slightly more general than the "offline" setting (also called *self-play*) considered by Chen et al. (2022); Xie et al. (2020), as we neither require the policy $\nu^t$ to be accessible to the max-player nor require both players to be directly controlled by a central controller.

**Additional Notations** With sequence-form representations, for any $\mu \in \Pi_{\max}$ and a sequence of functions $f = (f_h)_{h\in[H]}$ with $f_h : \mathcal{X}_h \times \mathcal{A} \to \mathbb{R}$, we let $\langle \mu, f \rangle := \sum_{h\in[H]} \sum_{x_h \in \mathcal{X}_h, a \in \mathcal{A}} \mu_{1:h}(x_h, a_h) f_h(x_h, a_h)$. We denote by $\mathcal{F}^t$ the $\sigma$-algebra generated by $\{(s_h^k, a_h^k, b_h^k, r_h^k)\}_{h\in[H], k\in[t]}$. For simplicity, we abbreviate $\mathbb{E}[\cdot \mid \mathcal{F}^t]$ as $\mathbb{E}^t[\cdot]$. The notation $\widetilde{\mathcal{O}}(\cdot)$ in this paper hides all the logarithmic factors.

---

[3] Our second algorithm can work in a more general case where the max-player only receives such features after the $t$-th episode ends.

# 3 LEAST-SQUARES ONLINE MIRROR DESCENT

In this section, we present our LSOMD algorithm, as well as its theoretical guarantees.

## 3.1 LINEAR LOSS ESTIMATOR

For a fixed $\nu^t$, Eq. (1) indicates that the value function $V^{\mu^t,\nu^t}$ is linear in $\mu^t$ (Kozuno et al., 2021):

$$V^{\mu^t,\nu^t} = \sum_{h=1}^{H} \sum_{(x_h,a_h)\in\mathcal{X}_h\times\mathcal{A}} \mu^t_{1:h}(x_h,a_h) \times \sum_{s_h\in x_h, b_h\in\mathcal{B}} p_{1:h}(s_h) \nu^t_{1:h}(y(s_h),b_h) \bar{r}_h(s_h,a_h,b_h) .$$

Hence, the regret in Eq. (3) can be rewritten as $\mathfrak{R}^T_{\max} = \max_{\mu^\dagger\in\Pi_{\max}} \sum_{t=1}^{T} \langle \mu^t - \mu^\dagger, \ell^t \rangle$, where we define the *loss function* in round $t$ as

$$\ell^t_h(x_h,a_h) := - \sum_{s_h\in x_h, b_h\in\mathcal{B}} p_{1:h}(s_h) \nu^t_{1:h}(y(s_h),b_h) \bar{r}_h(s_h,a_h,b_h) . \tag{4}$$

This implies that one can translate the regret minimization problem in Eq. (3) into a linear one.

To utilize the linear structure over the reward function to learn the unknown parameter $\boldsymbol{\theta}_h$, one may construct some sort of "linear" loss estimator $\hat{\boldsymbol{\theta}}_h$ of $\boldsymbol{\theta}_h$. However, this is more challenging in our case than it is in the case of linear bandits (Abbasi-Yadkori et al., 2011), linear MDPs (Jin et al., 2020), and linear perfect-information MGs (Xie et al., 2020), as we do not even know the underlying state $s_h$ and its associated feature vector $\boldsymbol{\phi}(s_h,a_h,b_h)$, making it impossible to regress $r_h(s_h,a_h,b_h)$ against $\boldsymbol{\phi}(s_h,a_h,b_h)$. To cope with this issue and build a least-squares loss estimator, we instead consider using the "feature vector" $\boldsymbol{\phi}(x_h,a_h)$ of $(x_h,a_h)$, which is a composite feature vector weighted by opponent's policy $\nu$ and transition:

$$\boldsymbol{\phi}^{\nu^t}(x_h,a_h) := - \sum_{(s_h,b_h)\in x_h\times\mathcal{B}} p_{1:h}(s_h)\nu^t_{1:h}(y(s_h),b_h)\boldsymbol{\phi}(s_h,a_h,b_h) , \tag{5}$$

which is assumed to be revealed to the max-player after the $t$-th episode ends in the offline setting as described in Section 2. Indeed, one can see that $\ell^t_h(x_h,a_h)$ is linear with $\boldsymbol{\phi}^{\nu^t}(x_h,a_h)$ and $\boldsymbol{\theta}_h$:

$$\ell^t_h(x_h,a_h) = \left\langle - \sum_{(s_h,b_h)\in x_h\times\mathcal{B}} p_{1:h}(s_h)\nu^t_{1:h}(y(s_h),b_h)\boldsymbol{\phi}(s_h,a_h,b_h), \boldsymbol{\theta}_h \right\rangle = \left\langle \boldsymbol{\phi}^{\nu^t}(x_h,a_h), \boldsymbol{\theta}_h \right\rangle .$$

Based on $\boldsymbol{\phi}^{\nu^t}(x_h,a_h)$, we further define the least-squares loss estimator $\hat{\boldsymbol{\theta}}_h$ as

$$\hat{\boldsymbol{\theta}}^t_h = \boldsymbol{Q}^{-1}_{\mu^t,h}\boldsymbol{\phi}^{\nu^t}(x_h,a_h)r_h(s_h,a_h,b_h) , \tag{6}$$

where $\boldsymbol{Q}_{\mu^t,h} = \sum_{(x_h,a_h)\in\mathcal{X}_h\times\mathcal{A}} \mu^t_{1:h}(x_h,a_h) \boldsymbol{\phi}^{\nu^t}(x_h,a_h)\boldsymbol{\phi}^{\nu^t}(x_h,a_h)^\top$ is the feature covariance matrix. Intuitively, this feature covariance matrix shares a similar spirit as its counterpart in the adversarial linear bandit literature (Lattimore & Szepesvári, 2020). However, we note that $\mu^t_{1:h}(\cdot,\cdot)$ here is not necessarily a distribution over $\mathcal{X}_h \times \mathcal{A}$.

This lemma shows $\hat{\boldsymbol{\theta}}^t_h$ is unbiased, which is critical in our analysis. See Appendix B.1 for its proof.

**Lemma 3.1.** *For any $t \in [T]$ and $h \in [H]$, it holds that $\mathbb{E}^{t-1}\left[\hat{\boldsymbol{\theta}}^t_h\right] = \boldsymbol{\theta}_h$.*

## 3.2 ALGORITHM DESCRIPTION

Our LSOMD algorithm follows the common scheme of OMD framework in that it runs OMD over $\Pi_{\max}$. Particularly, after interacting with the min-player using $\mu^t$, it computes the loss estimate $\hat{\ell}^t_h(x_h,a_h)$ with $\hat{\boldsymbol{\theta}}^t_h$ defined in Eq. (6) (Line 6 - Line 10). Subsequently, it updates the policy $\hat{\mu}^{t+1}$ by solving a regularized linear optimization problem:

$$\hat{\mu}^{t+1} = \arg\min_{\mu\in\Pi_{\max}} \eta \left\langle \mu, \hat{\ell}^t \right\rangle + D_\Psi(\mu\|\hat{\mu}^t) , \tag{7}$$

---

**Algorithm 1** `LSOMD` (max-player version)

1: **Input:** Tree-like structure of $\mathcal{X} \times \mathcal{A}$; Learning rate $\eta$.
2: **for** $t = 1$ to $T$ **do**
3:     **for** $h = 1$ to $H$ **do**
4:         Observe infoset $x_h^t$, execute $a_h^t \sim \mu_h^t(\cdot|x_h^t)$ and receive reward $r_h^t(s_h^t, a_h^t, b_h^t)$.
5:     **end for**
6:     **for** $h = 1$ to $H$ **do**
7:         Compute $\boldsymbol{Q}_{\mu^t, h} = \sum_{(x_h, a_h) \in \mathcal{X}_h \times \mathcal{A}} \mu_{1:h}^t(x_h, a_h) \boldsymbol{\phi}^{\nu^t}(x_h, a_h) \boldsymbol{\phi}^{\nu^t}(x_h, a_h)^\top$,
8:         Compute $\hat{\boldsymbol{\theta}}_h^t = \boldsymbol{Q}_{\mu^t, h}^{-1} \boldsymbol{\phi}^{\nu^t}(x_h, a_h) r_h(s_h, a_h, b_h)$,
9:     **end for**
10:    Construct loss estimate for all $(x_h, a_h)$ and $h \in [H]$: $\hat{\ell}_h^t(x_h, a_h) = \langle \boldsymbol{\phi}^{\nu^t}(x_h, a_h), \hat{\boldsymbol{\theta}}_h \rangle$.
11:    Receive composite features $\{\boldsymbol{\phi}^{\nu^{t+1}}(x, a)\}_{(x,a) \in \mathcal{X} \times \mathcal{A}}$.
12:    Update policy: $\mu^{t+1} = (1 - \gamma)\hat{\mu}^{t+1} + \gamma\pi$ with $\hat{\mu}^{t+1}$ computed in Eq. (7).
13: **end for**

---

where the potential function $\Psi$ is chosen as $\Psi(\mu) = \sum_{h=1}^H \sum_{(x_h, a_h) \in \mathcal{X}_h \times \mathcal{A}} \mu_{1:h}(x_h, a_h) \log \left( \frac{\mu_{1:h}(x_h, a_h)}{\sum_{a_h' \in \mathcal{A}} \mu_{1:h}(x_h, a_h')} \right)$. The induced Bregman divergence by $\Psi$ is typically called *dilated* entropy distance-generating function and is also adopted by Hoda et al. (2010b); Kroer et al. (2015b); Kozuno et al. (2021). Moreover, we note that the optimization problem in Eq. (7) can be efficiently solved via a backward update detailed in Appendix C.1, which turns out to be an extension of it in the tabular case considered by Kozuno et al. (2021) to our linear case.

However, there is one more caveat. In the analysis of `LSOMD`, it is required to control the variance of the loss estimates. To this end, at the end of episode $t$, after receiving composite feature vectors $\{\boldsymbol{\phi}^{\nu^{t+1}}(x, a)\}_{(x,a) \in \mathcal{X} \times \mathcal{A}}$, our `LSOMD` algorithm will compute $\hat{\mu}^{t+1}$ by solving Eq. (7) and then mix it with a uniform policy $\pi$, *i.e.*, $\mu^{t+1} = (1-\gamma)\hat{\mu}^{t+1} + \gamma\pi$ and $\pi(a \mid x) = 1/A$ for any $(x, a) \in \mathcal{X} \times \mathcal{A}$, where $\mu^{t+1}$ is the policy to be used in the next episode and $\gamma \in (0, 1)$ is the exploration parameter (Line 12).

### 3.3 ANALYSIS

Due to leveraging the feature vectors of infoset-actions, we additionally require the following assumption, which essentially guarantees that each direction of the feature space is well explored by the uniform policy $\pi$.

**Assumption 3.2.** *The uniform policy $\pi$ satisfies $\lambda_{\min}(\boldsymbol{Q}_{\pi, h}) \geq \rho > 0$, for any $h \in [H]$.*

The following theorem guarantees the regret upper bound of our `LSOMD` algorithm. Please see Appendix D for its proof.

**Theorem 3.3.** *In POMGs with linearly realizable rewards, by setting learning rate $\eta = \sqrt{\frac{\log A}{2TH(d+\rho^{-1})}}$ and exploration parameter $\gamma = \sqrt{\frac{X^2 \log A}{2HT(1+d\rho)\rho}}$, the regret bound of `LSOMD` in the offline setting is upper bounded by $\mathfrak{R}_{\max}^T \leq \mathcal{O}(\sqrt{(d + 1/\rho)HTX^2 \log A})$.*

**Remark 3.4.** *Compared with the regret upper bounds by Kozuno et al. (2021); Bai et al. (2022); Fiegel et al. (2023), the regret upper bound of our `LSOMD` does not have dependence on $A$, improves over Kozuno et al. (2021) by $\widetilde{\mathcal{O}}(\sqrt{HA})$ (omitting the dependence on $d$) but has an additional $\widetilde{\mathcal{O}}(\sqrt{HX})$ dependence compared with the minimax optimal result by Fiegel et al. (2023). On the other hand, as opposed to the high-probability regret guarantees in previous works studying tabular POMGs (Kozuno et al., 2021; Bai et al., 2022; Fiegel et al., 2023), the regret guarantee of our `LSOMD` algorithm only holds in expectation, which currently is not sufficient to be turned into an PAC algorithm for learning $\varepsilon$-NE. However, we would like to note again that this is the first line of algorithms that learns POMGs in the linear function approximation setting, with a regret guarantee independent of $A$. Also, we believe that it is possible to extend our results to high-probability results using self-concordant barrier potential functions and an increasing learning rate (Lee et al., 2020), which we leave as our future study.*

**Technique Overview** The proof of the regret upper bound of our `LSOMD` algorithm follows the common regret decomposition by bounding the *penalty* term and the *stability* term respectively. However, we note that bounding the stability term in our case is more difficult since bounding this term in the tabular case critically relies on the sparsity of the importance-weighted loss estimates, *i.e.*, the loss estimates are only non-zero at the experienced infoset-actions $\{(x_h^t, a_h^t)\}_{h \in [H]}$ (Kozuno et al., 2021). However, this does not apply in our case, where the linear loss estimator is utilized. To this end, we prove that the stability term in each episode $t$ is (approximately) bounded by the summation of the log-partition function $\log Z_1^t(x_1)$ for all $x_1 \in \mathcal{X}_1$. We then bound this term by relating it with the expectation of the inner product between $z^t$ and the loss estimate $\hat{\ell}^t$, in which $z_{1:h}^t(x_h, a_h)$ is the $a_h$-th value of the element-wise product of the random vectors independently sampled from categorical distributions specified by $\hat{\mu}^t$ along the path that leads to $x_h$ (*cf.*, Appendix D.2). Also, the solutions to the update for OMD in previous works (Kozuno et al., 2021; Bai et al., 2022) are tailored to the tabular case and do not go through our problem, which we addressed by devising an efficient update for the linear case (*cf.*, Appendix C.1).

## 4 Least-Squares Follow-the-Regularized-Leader

This section presents the other FTRL-based algorithm, termed as `LSFTRL`, and its regret guarantees.

### 4.1 Algorithm Description

Our second `LSFTRL` algorithm uses the same linear loss estimates as `LSOMD` (Line 7 - Line 12). To update the policy $\mu^{t+1}$ used at episode $t + 1$, it computes a linear optimization problem regularized by potential function $\{\Psi_h\}_{h \in [H}$ (Line 14):

$$\mu^{t+1} = \arg\min_{\mu \in \Pi_{\max}} \left\langle \mu, \hat{L}^t \right\rangle + \frac{1}{\eta} \sum_{h=1}^{H} \Psi_h \left( p_{1:h}^\star \cdot \mu_{1:h} \right), \tag{8}$$

where $\hat{L}^t = \sum_{k=1}^{t} \hat{\ell}^k$ is the cumulative loss estimate, $p_{1:h}^\star(x_h) = p_0^\star(x_1) \prod_{h'=1}^{h-1} p_{h'}^\star(x_{h'+1}|x_{h'}, a_{h'})$ with $p_h^\star(\cdot|x_h, a_h) \in \Delta_{C(x_h, a_h)}$ being a kind of "transition probability function" over $\mathcal{X}_h \times \mathcal{A} \times \mathcal{X}_{h+1}$, and $p_{1:h}^\star \cdot \mu_{1:h}$ is defined as $[p_{1:h}^\star \cdot \mu_{1:h}](x_h, a_h) = p_{1:h}^\star(x_h)\mu_{1:h}(x_h, a_h)$. Note that such $p^\star$ is well-defined due to the perfect recall condition, and $p_{1:h}^\star \cdot \mu_{1:h}$ is a probability distribution over the infoset-action pair $\mathcal{X}_h \times \mathcal{A}$ at step $h$. We also remark that similar approaches that combine the FTRL/OMD with $p_{1:h}^\star(\cdot)$ have also been exploited in previous works (*e.g.*, the balanced transition $p^\star$ of Bai et al. (2022); Fiegel et al. (2023) and the adversarial transition $p^{\star, \nu^t}$ of Fiegel et al. (2023)), but we will choose a different $p_{1:h}^\star(\cdot)$ satisfying

$$p^\star = \arg\max_{\tilde{p} \in \mathbb{P}^\star} \min_{h \in [H], x_h \in \mathcal{X}_h} \tilde{p}_{1:h}(x_h), \tag{9}$$

where $\mathbb{P}^\star$ denotes the set of all the valid transitions over infoset-actions. The computation of such $p^\star$ can be efficiently implemented using backward dynamic programming in $\mathcal{O}(XA)$ time, the details of which are postponed to Appendix E.2.3. As we shall see, the property of such $p^\star$ will serve as a key ingredient of the regret upper bound of our `LSFTRL` algorithm. Besides, `LSFTRL` chooses $\Psi_h(w_h) = \sum_{(x_h, a_h) \in \mathcal{X}_h \times \mathcal{A}} w_h(x_h, a_h) \log(w_h(x_h, a_h))$ as the negative entropy potential function (not to be confused with the dilated entropy potential function used in `LSOMD`). We also note that the computation of Eq. (8) can also be efficiently solved by reducing the update of `LSFTRL` to an OMD-like update, the details of which are deferred to Appendix C.3. The complete pseudo-code for `LSFTRL` algorithm is postponed to Appendix C.2.

### 4.2 Analysis

Let $p_{1:h}^\nu(x_h) = \sum_{s_h \in x_h} p_{1:h}(s_h) \nu_{1:h-1}(y(s_{h-1}), b_{h-1})$, which can be seen as the "probability" of reaching $x_h$ contributed by environment transition $\mathbb{P} = \{p_h\}_{h=0}^{H-1}$ and opponent's policy $\nu$. Similar to `LSOMD`, the regret upper bound of `LSFTRL` also depends on an exploratory assumption, detailed in the following. Please see Appendix E.1 for additional discussions on this assumption.

**Assumption 4.1.** *For any $t \in [T]$, $h \in [H]$ and $x_1, x_2 \in \mathcal{X}_h$, it holds that $p_{1:h}^{\nu^t}(x_1)/p_{1:h}^\star(x_2) \le \lambda$.*

We now present the regret upper bound of `LSFTRL`, with its proof postponed to Appendix E.2.

**Theorem 4.2.** *In POMGs with linearly realizable rewards, by setting learning rate $\eta = \sqrt{\frac{2\log(AX)}{Td\lambda}}$, the regret bound of* `LSFTRL` *in the offline setting is upper bounded by* $\mathfrak{R}_{\max}^T \leq \widetilde{\mathcal{O}}(\sqrt{H^2 d\lambda T})$.

**Remark 4.3.** `LSFTRL` *obtains the regret guarantee, which eliminates the dependence on both $X$ and $A$, in exchange for an exploratory assumption depending on the opponent's policy $\nu^t$. Compared with previous results, the regret upper bound in Theorem 4.2 improves over the minimax optimal regret $\widetilde{\mathcal{O}}(\sqrt{XAT})$ of Fiegel et al. (2023) by a factor $\widetilde{\mathcal{O}}(\sqrt{XA/H^2})$ (omitting the dependence on $d$ and $\lambda$). Note that if the max-player does not have access to $\lambda$, we can instead set $\eta = \sqrt{2\log(AX)/(Td)}$ without requiring the knowledge of $\lambda$, but at a slight cost of having the regret changing from $\widetilde{\mathcal{O}}(\sqrt{H^2 d\lambda T})$ to $\widetilde{\mathcal{O}}(\lambda\sqrt{H^2 dT})$. Besides, in cases where $\lambda$ is undesirably large (e.g., $\lambda \geq X/H$), a different choice of $p^\star$ by setting $p_{1:h}^\star(x_h) \equiv 1$ leads to the following regret guarantee of* `LSFTRL`.

**Theorem 4.4.** *In POMGs with linearly realizable rewards, by setting learning rate $\eta = \sqrt{\frac{2X\log A}{THd}}$ and $p_{1:h}^\star(x_h) \equiv 1$ for any $x_h \in \mathcal{X}$ and $h \in [H]$, the regret bound of* `LSFTRL` *in the offline setting is upper bounded by* $\mathfrak{R}_{\max}^T \leq \widetilde{\mathcal{O}}(\sqrt{HXdT})$.

**Remark 4.5.** *The proof of Theorem 4.4 is deferred to Appendix E.3. Note that by setting $p_{1:h}^\star(x_h) \equiv 1$, $p^\star$ is no longer a transition function over infoset-actions. Importantly, the regret in Theorem 4.4 improves over the minimax optimal result of Fiegel et al. (2023) by a factor $\widetilde{\mathcal{O}}(\sqrt{A/H})$ (omitting the dependence on $d$).*

**Technique Overview** We bound the regret of `LSFTRL` also by decomposing the regret into the penalty term and the stability term (Lattimore & Szepesvári, 2020), which is also adopted by Fiegel et al. (2023). However, we bound the stability term of `LSFTRL` with particular care such that the variances of the loss estimates are well-controlled by $\lambda$ in Assumption 4.1 and $d$ (*cf.*, Appendix E.2). Moreover, when bounding the penalty of `LSFTRL` with $p_{1:h}^\star(x_h) = 1$, we establish a refined analysis that shaves off an $\mathcal{O}(\sqrt{A})$ factor, compared with the direct combination of the original analysis of Fiegel et al. (2023) and the setting of $p_{1:h}^\star(x_h) = 1$ (*cf.*, Appendix E.3).

### 4.3 Regret Lower Bound

We also provide a regret lower bound of learning POMGs with linearly realizable rewards in the following theorem, the proof of which is deferred to Appendix F.

**Theorem 4.6.** *Suppose $A \geq 2$, $d \geq 2$ and $T \geq 2d^2$. Then for any algorithm* Alg *that controls the max-player, generates and executes policies $\{\mu^t\}_{t\in[T]}$, there exists an POMG instance on which $\mathfrak{R}_{\max}^T \geq \Omega(\sqrt{d\min(d,H)T})$.*

**Remark 4.7.** *We conjecture that the regret lower bound can be further improved to $\mathfrak{R}_{\max}^T \geq \Omega(\sqrt{dHT})$, and currently our regret upper bounds of* `LSOMD` *and* `LSFTRL` *with the second initialization are loose by $\widetilde{\mathcal{O}}(X)$ and $\widetilde{\mathcal{O}}(\sqrt{X})$ factors and regret upper bound of* `LSFTRL` *with the first initialization is loose by an $\widetilde{\mathcal{O}}(\sqrt{H})$ factor (omitting the dependence on $\rho$ and $\lambda$). We leave the investigation into the possible improvements of the upper and lower bounds as our future studies.*

## 5 Conclusion

In this work, we make the first step towards provably efficient learning of the two-player, zero-sum IIEFGs with linear function approximation, in the formulation of POMGs with linearly realizable rewards and unknown transitions. It is proven that, the proposed `LSOMD` algorithm obtains an $\widetilde{\mathcal{O}}(\sqrt{(d + 1/\rho)HX^2T})$ regret, and the `LSFTRL` algorithm attains regret of orders $\widetilde{\mathcal{O}}(\sqrt{H^2 d\lambda T})$ and $\widetilde{\mathcal{O}}(\sqrt{HXdT})$. We accomplish this by devising the first least-squares loss estimator for this setting, along with new ingredients in the analysis for both the `LSOMD` and `LSFTRL` algorithms, which may be of independent interest. Also, we provide an $\Omega(\sqrt{d\min(d,H)T})$ regret lower bound. Besides, there are also several interesting future directions to be explored. One natural question might be how to obtain high-probability results in this challenging problem so as to find an $\varepsilon$-NE. The other question might be whether it is possible generalize the proposed algorithms and results to multi-player general-sum POMGs. We hope our results may shed light on better understandings of learning large-scale POMGs and we leave these extensions as our further studies.

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
