## A  PROPERTIES OF THE GAME

The lemma below delineates the key property of $p^\star$ as transition probability functions.

**Lemma A.1.** *For any $h \in [H]$, any $p^\star$ as transition probability function over infoset-actions and any policy $\mu \in \Pi_{\max}$ of the max-player, it holds that*

$$\sum_{(x_h, a_h) \in \mathcal{X}_h \times \mathcal{A}} p^\star_{1:h}(x_h) \mu_{1:h}(x_h, a_h) = 1 \,.$$

*Proof.* By the definition of perfect recall and transition probability functions over infoset-actions, we have

$$\begin{aligned}
\mathbb{P}^{\mu,\nu}(x_h, a_h) &= \mathbb{P}^{\mu,\nu}(x_1, \ldots, x_h, a_h) \\
&= p^\star_0(x_1) \prod_{h'=1}^{h-1} p^\star_{h'}(x_{h'+1} | x_{h'}, a_{h'}) \cdot \prod_{h'=1}^{h} \mu_{h'}(a_{h'} | x_{h'}) \\
&= p^\star(x_h) \mu_{1:h}(x_h, a_h) \,.
\end{aligned}$$

The proof is thus concluded by noticing that $\sum_{(x_h, a_h) \in \mathcal{X}_h \times \mathcal{A}} \mathbb{P}^{\mu,\nu}(x_h, a_h) = 1$.  □

## B  PROPERTIES OF THE LEAST-SQUARES LOSS ESTIMATOR

This section presents the proofs of two key properties of the proposed least-squares loss estimator.

### B.1  UNBIASNESS OF THE LOSS ESTIMATOR

*Proof of Lemma 3.1.*

$$\begin{aligned}
\mathbb{E}^{t-1}\left[\hat{\boldsymbol{\theta}}_h^t\right] &= \mathbb{E}^{\mu^t, \nu^t}\left[\hat{\boldsymbol{\theta}}_h^t\right] \\
&= \mathbb{E}^{\mu^t, \nu^t}\left[\boldsymbol{Q}_{\mu^t, h}^{-1} \cdot \boldsymbol{\phi}^{\nu^t}(x_h, a_h) \cdot r_h(s_h, a_h, b_h)\right] \\
&= \boldsymbol{Q}_{\mu^t, h}^{-1} \sum_{x_h \in \mathcal{X}_h} \sum_{s_h \in x_h} \sum_{a_h \in \mathcal{A}} \sum_{b_h \in \mathcal{B}} \mathbb{P}^{\mu^t, \nu^t}(s_h, a_h, b_h) \boldsymbol{\phi}^{\nu^t}(x_h, a_h) \bar{r}_h(s_h, a_h, b_h) \\
&= \boldsymbol{Q}_{\mu^t, h}^{-1} \sum_{x_h \in \mathcal{X}_h} \sum_{a_h \in \mathcal{A}} \mu_{1:h}^t(x_h, a_h) \boldsymbol{\phi}^{\nu^t}(x_h, a_h) \sum_{s_h \in x_h} \sum_{b_h \in \mathcal{B}} p_{1:h}(s_h) \nu_{1:h}^t(y(s_h), b_h) \bar{r}_h(s_h, a_h, b_h) \\
&= \boldsymbol{Q}_{\mu^t, h}^{-1} \sum_{x_h \in \mathcal{X}_h} \sum_{a_h \in \mathcal{A}} \mu_{1:h}^t(x_h, a_h) \boldsymbol{\phi}^{\nu^t}(x_h, a_h) \left\langle \boldsymbol{\phi}^{\nu^t}(x_h, a_h), \boldsymbol{\theta}_h \right\rangle \\
&= \boldsymbol{Q}_{\mu^t, h}^{-1} \left( \sum_{x_h \in \mathcal{X}_h} \sum_{a_h \in \mathcal{A}} \mu_{1:h}^t(x_h, a_h) \boldsymbol{\phi}^{\nu^t}(x_h, a_h) \boldsymbol{\phi}^{\nu^t}(x_h, a_h)^\top \right) \boldsymbol{\theta}_h \\
&= \boldsymbol{\theta}_h \,.
\end{aligned}$$

□

### B.2  VARIANCE OF THE LOSS ESTIMATOR

Importantly, the following lemma shows that the "variance" of the proposed loss estimator is well controlled.

**Lemma B.1.** *For any $h \in [H]$, it holds that*

$$\mathbb{E}^{t-1}\left[ \sum_{(x_h, a_h) \in \mathcal{X}_h \times \mathcal{A}} \mu_{1:h}^t(x_h, a_h) \hat{\ell}_h^t(x_h, a_h)^2 \right] \leq d \,. \tag{10}$$

*Proof.*

$$\mathbb{E}^{t-1}\left[\sum_{(x_h,a_h)\in\mathcal{X}_h\times\mathcal{A}}\mu_{1:h}^t(x_h,a_h)\hat{\ell}_h^t(x_h,a_h)^2\right]$$

$$=\sum_{(x_h,a_h)\in\mathcal{X}_h\times\mathcal{A}}\mu_{1:h}^t(x_h,a_h)\phi^{\nu^t}(x_h,a_h)^\top\mathbb{E}^{\mu^t,\nu^t}\left[\hat{\boldsymbol{\theta}}_h^t\left(\hat{\boldsymbol{\theta}}_h^t\right)^\top\right]\phi^{\nu^t}(x_h,a_h)$$

$$=\sum_{(x_h,a_h)\in\mathcal{X}_h\times\mathcal{A}}\mu_{1:h}^t(x_h,a_h)\phi^{\nu^t}(x_h,a_h)^\top$$

$$\cdot\mathbb{E}^{\mu^t,\nu^t}\left[r_h(s_h,a_h,b_h)^2\boldsymbol{Q}_{\mu^t,h}^{-1}\phi^{\nu^t}(x_h,a_h)\phi^{\nu^t}(x_h,a_h)^\top\boldsymbol{Q}_{\mu^t,h}^{-1}\right]\phi^{\nu^t}(x_h,a_h)$$

$$\leq\sum_{(x_h,a_h)\in\mathcal{X}_h\times\mathcal{A}}\mu_{1:h}^t(x_h,a_h)\phi^{\nu^t}(x_h,a_h)^\top\mathbb{E}^{\mu^t,\nu^t}\left[\boldsymbol{Q}_{\mu^t,h}^{-1}\phi^{\nu^t}(x_h,a_h)\phi^{\nu^t}(x_h,a_h)^\top\boldsymbol{Q}_{\mu^t,h}^{-1}\right]\phi^{\nu^t}(x_h,a_h)$$

$$=\sum_{(x_h,a_h)\in\mathcal{X}_h\times\mathcal{A}}\mu_{1:h}^t(x_h,a_h)\phi^{\nu^t}(x_h,a_h)^\top\boldsymbol{Q}_{\mu^t,h}^{-1}$$

$$\cdot\left(\sum_{(x_h,a_h)\in\mathcal{X}_h\times\mathcal{A}}p^{\nu^t}(x_h)\mu_{1:h}^t(x_h,a_h)\phi^{\nu^t}(x_h,a_h)\phi^{\nu^t}(x_h,a_h)^\top\right)\boldsymbol{Q}_{\mu^t,h}^{-1}\phi^{\nu^t}(x_h,a_h)$$

$$=\operatorname{tr}\left[\left(\sum_{(x_h,a_h)\in\mathcal{X}_h\times\mathcal{A}}\mu_{1:h}^t(x_h,a_h)\phi^{\nu^t}(x_h,a_h)\phi^{\nu^t}(x_h,a_h)^\top\boldsymbol{Q}_{\mu^t,h}^{-1}\right)\right.$$

$$\left.\cdot\left(\sum_{(x_h,a_h)\in\mathcal{X}_h\times\mathcal{A}}p^{\nu^t}(x_h)\mu_{1:h}^t(x_h,a_h)\phi^{\nu^t}(x_h,a_h)\phi^{\nu^t}(x_h,a_h)^\top\boldsymbol{Q}_{\mu^t,h}^{-1}\right)\right]$$

$$=\operatorname{tr}\left[\boldsymbol{I}_d\cdot\sum_{(x_h,a_h)\in\mathcal{X}_h\times\mathcal{A}}p^{\nu^t}(x_h)\mu_{1:h}^t(x_h,a_h)\phi^{\nu^t}(x_h,a_h)\phi^{\nu^t}(x_h,a_h)^\top\boldsymbol{Q}_{\mu^t,h}^{-1}\right]$$

$$\leq\operatorname{tr}\left[\sum_{(x_h,a_h)\in\mathcal{X}_h\times\mathcal{A}}\mu_{1:h}^t(x_h,a_h)\phi^{\nu^t}(x_h,a_h)\phi^{\nu^t}(x_h,a_h)^\top\boldsymbol{Q}_{\mu^t,h}^{-1}\right]$$

$$=\operatorname{tr}\left[\boldsymbol{I}_d\right]=d.$$

$\square$

## C COMPUTATION ISSUE

In this section, we present efficient solutions to the optimization problems of both `LSOMD` and `LSFTRL`.

### C.1 EFFICIENT UPDATE FOR `LSOMD`

To begin with, we first introduce a generalized version of OMD update in Eq. (7), which leverages learning rates adaptive to each infoset. Specifically, given any list of learning rates $\eta :=(\eta_h(x_h))_{h\in[H],x_h\in\mathcal{X}}$, the potential function is defined as

$$\Psi_\eta(\mu)=\sum_{h=1}^H\sum_{(x_h,a_h)\in\mathcal{X}_h\times\mathcal{A}}\frac{\mu_{1:h}(x_h,a_h)}{\eta_h(x_h)}\log\left(\frac{\mu_{1:h}(x_h,a_h)}{\sum_{a_h'\in\mathcal{A}}\mu_{1:h}(x_h,a_h')}\right).$$

By the fact that for all positive $\mu\in\Pi_{\max}$, the derivative of $\Psi_\eta(\mu)$ satisfies

$$\nabla_{x_h,a_h}\Psi_\eta(\mu)=\frac{1}{\eta_h(x_h)}\log(\mu_h(a_h|x_h)),$$

one can see that $\Psi_\eta(\mu)$ induces the dilated distance generating function

$$D_{\Psi_\eta}(\mu^1\|\mu^2) = \sum_{h=1}^{H} \sum_{(x_h,a_h)\in\mathcal{X}_h\times\mathcal{A}} \frac{\mu^1_{1:h}(x_h,a_h)}{\eta_h(x_h)} \log\frac{\mu^1_h(a_h|x_h)}{\mu^2_h(a_h|x_h)}.$$

The generalized version of OMD update in Eq. (7) is given as follows:

$$\mu^{t+1} = \arg\min_{\mu\in\Pi_{\max}} \left\langle \mu, \hat{\ell}^t \right\rangle + D_{\Psi_\eta}(\mu\|\mu^t)$$

$$= \arg\min_{\mu\in\Pi_{\max}} \left\langle \mu, \hat{\ell}^t \right\rangle + \sum_{h=1}^{H} \sum_{(x_h,a_h)\in\mathcal{X}_h\times\mathcal{A}} \frac{\mu_{1:h}(x_h,a_h)}{\eta_h(x_h)} \log\frac{\mu_h(a_h|x_h)}{\mu^t_h(a_h|x_h)}. \quad (11)$$

We remark that $\eta := (\eta_h(x_h))_{h\in[H],x_h\in\mathcal{X}}$ also generalizes the of *balanced transitions* used in Farina et al. (2020); Bai et al. (2022); Fiegel et al. (2023).

The solution to Eq. (11) is given in the following proposition. Notice that the solution to this optimization problem of previous works (Kozuno et al., 2021; Bai et al., 2022; Fiegel et al., 2023) critically relies on the sparsity of their importance-weighted loss estimator, which only permits non-zero loss estimates along the experienced trajectory. However, our solution supports the loss estimator with non-zero loss estimates for arbitrary infoset-action pairs.

**Proposition C.1.** *The solution to the update rule in Eq. (11) is as followed:*

$$\mu^{t+1}_h(a_h|x_h) = \mu^t_h(a_h|x_h)\exp\left\{-\eta_h(x_h)\hat{\ell}^t_h(x_h,a_h) + \sum_{x_{h+1}\in C(x_h,a_h)} \frac{\eta_h(x_h)}{\eta_{h+1}(x_{h+1})}\log Z^t_{h+1}(x_{h+1}) - \log Z^t_h(x_h)\right\},$$

*where*

$$Z^t_h(x_h) = \sum_{a_h\in\mathcal{A}} \mu^t_h(a_h|x_h)\exp\left\{-\eta_h(x_h)\hat{\ell}^t_h(x_h,a_h) + \sum_{x_{h+1}\in C(x_h,a_h)} \frac{\eta_h(x_h)}{\eta_{h+1}(x_{h+1})}\log Z^t_{h+1}(x_{h+1})\right\},$$

*and for notational convenience, we define that $\forall x_H\in X_H$, it has a unique descendant $x_{H+1}$ such that $Z^t_{H+1}(x_{H+1})=1$.*

*Proof.* We first note that

$$\left\langle \mu, \hat{\ell}^t \right\rangle + D_{\Psi_\eta}(\mu\|\mu^t)$$

$$= \sum_{h=1}^{H} \sum_{(x_h,a_h)\in\mathcal{X}_h\times\mathcal{A}} \mu_{1:h}(x_h,a_h)\left[\hat{\ell}^t_h(x_h,a_h) + \frac{1}{\eta_h(x_h)}\log\frac{\mu_h(a_h|x_h)}{\mu^t_h(a_h|x_h)}\right]$$

$$= \sum_{h=1}^{H} \sum_{x_h\in\mathcal{X}_h} \mu_{1:h-1}(x_h)\left[\left\langle \mu_h(\cdot|x_h), \hat{\ell}^t_h(x_h,\cdot)\right\rangle + \frac{D_{\mathrm{KL}}\left(\mu_h(\cdot|x_h)\|\mu^t_h(\cdot|x_h)\right)}{\eta_h(x_h)}\right]. \quad (12)$$

We now prove the proposition through backward induction over $h = H, \ldots, 1$. For $h = H, x_H \in \mathcal{X}_H$, it is easy to see that

$$\mu^{t+1}_H(a_H|x_H) \propto_{a_H} \mu^t_H(a_H|x_H)\exp\left\{-\eta_h(x_h)\hat{\ell}^t_H(x_H,a_H)\right\}$$

$$= \mu^t_H(a_H|x_H)\exp\left\{-\eta_h(x_h)\hat{\ell}^t_H(x_H,a_H) - \log Z^t_H(x_H)\right\},$$

where $Z^t_H(x_H) = \sum_{a_H\in\mathcal{A}} \mu^t_H(a_H|x_H)\exp\left\{-\eta_h(x_h)\hat{\ell}^t_H(x_H,a_H)\right\} > 0$ is a normalization factor.

Suppose the proposition holds from step $h+1$ to $H$ and consider the $h$-th step. Substituting the induction hypothesis, one can see that Eq. (12) can be expressed as follows:

$$\sum_{h'=1}^{H} \sum_{(x_{h'},a_{h'}) \in \mathcal{X}_{h'} \times \mathcal{A}} \mu_{1:h'}(x_{h'},a_{h'}) \left[ \hat{\ell}_{h'}^{t}(x_{h'},a_{h'}) + \frac{1}{\eta_{h'}(x_{h'})} \log \frac{\mu_{h'}(a_{h'}|x_{h'})}{\mu_{h'}^{t}(a_{h'}|x_{h'})} \right]$$

$$= \sum_{h'=1}^{H} \sum_{x_{h'} \in \mathcal{X}_{h'}} \mu_{1:h'-1}(x_{h'}) \left[ \left\langle \mu_{h'}(\cdot|x_{h'}), \hat{\ell}_{h'}^{t}(x_{h'},\cdot) \right\rangle + \frac{D_{\mathrm{KL}}\left(\mu_{h'}(\cdot|x_{h'})||\mu_{h'}^{t}(\cdot|x_{h'})\right)}{\eta_{h'}(x_{h'})} \right]$$

$$= \sum_{h'=1}^{h} \sum_{x_{h'} \in \mathcal{X}_{h'}} \mu_{1:h'-1}(x_{h'}) \left[ \left\langle \mu_{h'}(\cdot|x_{h'}), \hat{\ell}_{h'}^{t}(x_{h'},\cdot) \right\rangle + \frac{D_{\mathrm{KL}}\left(\mu_{h'}(\cdot|x_{h'})||\mu_{h'}^{t}(\cdot|x_{h'})\right)}{\eta_{h'}(x_{h'})} \right]$$

$$+ \sum_{h'=h+1}^{H} \left[ \sum_{x_{h'+1} \in \mathcal{X}_{h'+1}} \frac{\mu_{1:h'}(x_{h'+1})}{\eta_{h'+1}(x_{h'+1})} \log Z_{h'+1}^{t}(x_{h'+1}) - \sum_{x_{h'} \in \mathcal{X}_{h'}} \frac{\mu_{1:h'-1}(x_{h'})}{\eta_{h'}(x_{h'})} \log Z_{h'}^{t}(x_{h'}) \right]$$

$$= \sum_{h'=1}^{h} \sum_{x_{h'} \in \mathcal{X}_{h'}} \mu_{1:h'-1}(x_{h'}) \left[ \left\langle \mu_{h'}(\cdot|x_{h'}), \hat{\ell}_{h'}^{t}(x_{h'},\cdot) \right\rangle + \frac{D_{\mathrm{KL}}\left(\mu_{h'}(\cdot|x_{h'})||\mu_{h'}^{t}(\cdot|x_{h'})\right)}{\eta_{h'}(x_{h'})} \right]$$

$$- \sum_{x_{h+1} \in \mathcal{X}_{h+1}} \frac{\mu_{1:h}(x_{h+1})}{\eta_{h+1}(x_{h+1})} \log Z_{h+1}^{t}(x_{h+1})$$

$$= \sum_{h'=1}^{h-1} \sum_{x_{h'} \in \mathcal{X}_{h'}} \mu_{1:h'-1}(x_{h'}) \left[ \left\langle \mu_{h'}(\cdot|x_{h'}), \hat{\ell}_{h'}^{t}(x_{h'},\cdot) \right\rangle + \frac{D_{\mathrm{KL}}\left(\mu_{h'}(\cdot|x_{h'})||\mu_{h'}^{t}(\cdot|x_{h'})\right)}{\eta_{h'}(x_{h'})} \right]$$

$$+ \sum_{x_h \in \mathcal{X}_h} \mu_{1:h-1}(x_h) \left[ \underbrace{\left\langle \mu_h(\cdot|x_h), \hat{\ell}_h^{t}(x_h,\cdot) - \sum_{x_{h+1} \in C(x_h,\cdot)} \frac{\log Z_{h+1}^{t}(x_{h+1})}{\eta_{h+1}(x_{h+1})} \right\rangle + \frac{D_{\mathrm{KL}}\left(\mu_h(\cdot|x_h)||\mu_h^{t}(\cdot|x_h)\right)}{\eta_h(x_h)}}_{\heartsuit} \right].$$

By optimizing $(\heartsuit)$, one can derive that

$$\mu_h^{t+1}(a_h|x_h) = \mu_h^{t}(a_h|x_h) \exp\left\{ -\eta_h(x_h)\hat{\ell}_h^{t}(x_h,a_h) + \sum_{x_{h+1} \in C(x_h,a_h)} \frac{\eta_h(x_h)}{\eta_{h+1}(x_{h+1})} \log Z_{h+1}^{t}(x_{h+1}) - \log Z_h^{t}(x_h) \right\},$$

$$Z_h^{t}(x_h) = \sum_{a_h \in \mathcal{A}} \mu_h^{t}(a_h|x_h) \exp\left\{ -\eta_h(x_h)\hat{\ell}_h^{t}(x_h,a_h) + \sum_{x_{h+1} \in C(x_h,a_h)} \frac{\eta_h(x_h)}{\eta_{h+1}(x_{h+1})} \log Z_{h+1}^{t}(x_{h+1}) \right\},$$

which thus concludes the proof. $\square$

Proposition C.1 immediately implies the efficient update procedure for LSOMD, detailed in Algorithm 2, by setting $\eta_h(x_h) \equiv \eta$ for all $x_h \in \mathcal{X}$ in Proposition C.1. In what follows, for notational convenience, we denote $J_h^{t}(x_h,a_h) = -\eta_h(x_h)\hat{\ell}_h^{t}(x_h,a_h) + \sum_{x_{h+1} \in C(x_h,a_h)} \frac{\eta_h(x_h)}{\eta_{h+1}(x_{h+1})} \log Z_{h+1}^{t}(x_{h+1})$ as the surrogate loss.

## C.2 LSFTRL Algorithm

## C.3 Efficient Update for LSFTRL

To solve the update of LSFTRL, we follow the same idea as Fiegel et al. (2023) that translating the update of FTRL into the update of OMD-like update. In specific, the Proposition F.2 of Fiegel et al. (2023) shows that the update of Eq. (8) is equivalent to the solution to the following optimization problem:

$$\mu^{t} = \operatorname*{arg\,min}_{\mu \in \Pi_{\max}} \left\langle \mu, \hat{L}^{t} \right\rangle + D_{\eta^{\star}}(\mu, \mu^{\star}), \tag{13}$$

---

**Algorithm 2** `Update-of-LSOMD`

---

1: **Input:** Tree-like structure of $\mathcal{X} \times \mathcal{A}$, $\hat{\mu}^t$ given by update Eq. (7); fixed learning rates $\eta$; the loss estimates $\left\{ \hat{\ell}_h^t(x_h, a_h) \right\}_{(x_h, a_h) \in \mathcal{X} \times \mathcal{A}}$.
2: **Initialization:** For all $x_H$ in $\mathcal{X}_H$, initialize $Z^t(x_{H+1}) = 1$.
3: **for** $h = H$ to 1 **do**
4:     **for** $x_h$ in $\mathcal{X}_h$ **do**
5:         Compute $J_h^t(x_h, a_h) = -\eta \hat{\ell}_h^t(x_h, a_h) + \sum_{x_{h+1} \in C(x_h, a_h)} \log Z_{h+1}^t(x_{h+1})$,
6:         Compute $Z_h^t(x_h) = \sum_{a_h \in \mathcal{A}} \hat{\mu}_h^t(a_h | x_h) \exp \left( J_h^t(x_h, a_h) \right)$,
7:         **for** $a_h$ in $\mathcal{A}$ **do**
8:             Compute $\hat{\mu}_h^{t+1}(a_h | x_h) = \hat{\mu}_h^t(a_h | x_h) \exp \left( J_h^t(x_h, a_h) - \log Z_h^t(x_h) \right)$.
9:         **end for**
10:    **end for**
11: **end for**

---

**Algorithm 3** `LSFTRL` (max-player version)

---

1: **Input:** Tree-like structure of $\mathcal{X} \times \mathcal{A}$; Learning rates $\eta$; $p^\star$.
2: **for** $t = 1$ to $T$ **do**
3:     **for** $h = 1$ to $H$ **do**
4:         Observe infoset $x_h^t$.
5:         Execute $a_h^t \sim \mu_h^t(\cdot | x_h^t)$ and receive reward $r_h^t(s_h^t, a_h^t, b_h^t)$.
6:     **end for**
7:     Receive composite features $\left\{ \phi^{\nu^t}(x, a) \right\}_{(x, a) \in \mathcal{X} \times \mathcal{A}}$.
8:     **for** $h = 1$ to $H$ **do**
9:         Compute $\boldsymbol{Q}_{\mu^t, h} = \sum_{(x_h, a_h) \in \mathcal{X}_h \times \mathcal{A}} \mu_{1:h}^t(x_h, a_h) \phi^{\nu^t}(x_h, a_h) \phi^{\nu^t}(x_h, a_h)^\top$,
10:        Compute $\hat{\boldsymbol{\theta}}_h^t = \boldsymbol{Q}_{\mu^t, h}^{-1} \phi^{\nu^t}(x_h, a_h) r_h(s_h, a_h, b_h)$,
11:     **end for**
12:     Construct loss estimate for all $(x_h, a_h)$ and $h \in [H]$: $\hat{\ell}_h^t(x_h, a_h) = \left\langle \phi^{\nu^t}(x_h, a_h), \hat{\boldsymbol{\theta}}_h \right\rangle$.
13:     Compute cumulative loss estimate at episode $t$: $\hat{L}^t = \hat{L}^{t-1} + \hat{\ell}^t$.
14:     Compute update (8) using `Update-of-LSFTRL`.
15: **end for**

---

where $\eta^\star := (\eta_h^\star(x_h))_{h, x_h}$ is a learning rate adaptive to each infoset, $\mu^\star$ is a base policy and we define $D_{\eta^\star}\left(\mu^1, \mu^0\right) := \sum_{h=1}^H \sum_{(x_h, a_h) \in \mathcal{A}(\mathcal{X}_h)} \frac{\mu_{1:h}^1(x_h, a_h)}{\eta_h^\star(x_h)} \log \frac{\mu_h^1(a_h | x_h)}{\mu_h^0(a_h | x_h)}$.

Therefore, to solve Eq. (8), for all $x_h \in \mathcal{X}$, we first set the adaptive learning rate $\eta^\star$ as

$$\eta_h^\star(x_h) = \frac{\eta}{(H - h + 1) p_{1:h}^\star(x_h)}, \tag{14}$$

and set the base policy $\mu^\star$ as

$$\mu^\star = \underset{\mu^\star \in \Pi_{\max}}{\arg \min} \sum_{h=1}^H \Psi_h \left( p_{1:h}^\star \cdot \mu_{1:h}^\star \right), \tag{15}$$

which can be computed efficiently via backward dynamic programming in $\mathcal{O}(XA)$ time. Then, combined with the efficient update procedure of `LSOMD` in Algorithm 2, the solution to the update of `LSFTRL` can be obtained by substituting $\mu^t$ with $\mu^\star$, the details of which are presented in Algorithm 4 for completeness.

## D   PROOF OF REGRET GUARANTEE OF `LSOMD`

In this section, we present the proof of the regret guarantee of `LSOMD`.

---

**Algorithm 4** `Update-of-LSFTRL`

---

1: **Input:** Tree-like structure of $\mathcal{X} \times \mathcal{A}$ ; fixed learning rates $\eta$ ; transition probability function $p^\star$ ;
 cumulative loss estimates $\left\{ \hat{L}_h^t(x_h, a_h) \right\}_{(x_h, a_h) \in \mathcal{X} \times \mathcal{A}}$.

2: **Initialization:** For all $x_H$ in $\mathcal{X}_H$, initialize $Z^t(x_{H+1}) = 1$ ; Set adaptive learning rates $\eta^\star$
 according to Eq. (14) ; Set base policy $\mu^\star$ according to Eq. (15) .

3: **for** $h = H$ to $1$ **do**
4:   **for** $x_h$ in $\mathcal{X}_h$ **do**
5:    Compute $J_h^t(x_h, a_h) = -\eta_h^\star(x_h)\hat{L}^t(x_h, a_h) + \sum_{x_{h+1} \in C(x_h, a_h)} \frac{\eta_h^\star(x_h)}{\eta_{h+1}^\star(x_{h+1})} \log Z_{h+1}^t(x_{h+1})$,

6:    Compute $Z_h^t(x_h) = \sum_{a_h \in \mathcal{A}} \mu_h^\star(a_h|x_h) \exp\left(J_h^t(x_h, a_h)\right)$,
7:    **for** $a_h$ in $\mathcal{A}$ **do**
8:     Compute $\mu_h^{t+1}(a_h|x_h) = \mu_h^\star(a_h|x_h) \exp\left(J_h^t(x_h, a_h) - \log Z_h^t(x_h)\right)$.
9:    **end for**
10:   **end for**
11: **end for**

---

*Proof of Theorem 3.3.* First note that

$$\left\langle \hat{\mu}^t - \mu^\dagger, \hat{\ell}^t \right\rangle = D_\eta(\hat{\mu}^t \| \hat{\mu}^{t+1}) - D_\eta(\hat{\mu}^t \| \hat{\mu}^t) - (D_\eta(\mu^\dagger \| \hat{\mu}^{t+1}) - D_\eta(\mu^\dagger \| \hat{\mu}^t))$$
$$= D_\eta(\mu^\dagger \| \hat{\mu}^t) - D_\eta(\mu^\dagger \| \hat{\mu}^{t+1}) + D_\eta(\hat{\mu}^t \| \hat{\mu}^{t+1}) .$$

Taking summation of the above display over $t$ and telescoping the sum, we have

$$\sum_{t=1}^{T} \left\langle \hat{\mu}^t - \mu^\dagger, \hat{\ell}^t \right\rangle \leq \underbrace{D_\eta(\mu^\dagger \| \hat{\mu}^1)}_{\text{PENALTY}} + \underbrace{\sum_{t=1}^{T} D_\eta(\hat{\mu}^t \| \hat{\mu}^{t+1})}_{\text{STABILITY}} . \tag{16}$$

On the other hand, by the unbiasedness of $\hat{\ell}^t$ and the tower rule, it holds that

$$\mathbb{E}\left[\langle \mu^t - \mu^\dagger, \ell^t \rangle\right] = \mathbb{E}\left[\mathbb{E}^{t-1}\left[\left\langle \mu^t - \mu^\dagger, \hat{\ell}^t \right\rangle\right]\right] = \mathbb{E}\left[\left\langle \mu^t - \mu^\dagger, \hat{\ell}^t \right\rangle\right] , \tag{17}$$

where recall that $\mu^t = (1 - \gamma)\hat{\mu}^t + \gamma\pi^t$.

Combining Eq. (16) and Eq. (17), along with the definition of regret in Eq. (3), one can deduce that

$$\mathfrak{R}_{\max}^T \leq \max_{\mu^\dagger \in \Pi_{\max}} (1 - \gamma)\mathbb{E}\left[\underbrace{D_\eta(\mu^\dagger \| \hat{\mu}^1)}_{\text{PENALTY}} + \underbrace{\sum_{t=1}^{T} D_\eta(\hat{\mu}^t \| \hat{\mu}^{t+1})}_{\text{STABILITY}}\right] + 2\gamma HT$$

$$\leq (1 - \gamma)\left(\frac{X \log A}{\eta} + \eta TXHd\right) + 2\gamma HT$$

$$\leq \left(\frac{X \log A}{\eta} + \eta TXHd\right) + 2\eta HXTd\alpha^{-1}$$

$$= \frac{X \log A}{\eta} + 2\eta TXHd(1 + \alpha^{-1}) ,$$

where the second inequality comes from Lemma D.1 and D.5.

Finally, the proof is concluded by substituting $\eta = \sqrt{\frac{\log A}{2THd(1+\alpha^{-1})}}$ and $\gamma = \sqrt{\frac{Xd \log A\alpha^{-1}}{2HT(1+\alpha^{-1})}}$. $\qquad\square$

In our `LSOMD`, we set the learning rate to be constant, *i.e.*, $\eta_h(x_h) \equiv \eta$ for all $x_h \in \mathcal{X}$.

## D.1 BOUNDING THE PENALTY TERM

**Lemma D.1.** *The* PENALTY *term is bounded by*

$$\text{PENALTY} \leq \frac{X \log A}{\eta}.$$

*Proof.*

$$
\begin{aligned}
D_\eta(\mu^\dagger \| \hat{\mu}^1) &= \sum_{h=1}^{H} \sum_{(x_h, a_h) \in \mathcal{X}_h \times \mathcal{A}} \frac{\mu^\dagger_{1:h}(x_h, a_h)}{\eta} \log \frac{\mu^\dagger_h(a_h | x_h)}{\hat{\mu}^1_h(a_h | x_h)} \\
&\leq \sum_{h=1}^{H} \sum_{(x_h, a_h) \in \mathcal{X}_h \times \mathcal{A}} \frac{\mu^\dagger_{1:h}(x_h, a_h)}{\eta} \log \hat{\mu}^1_h(a_h | x_h) \\
&= \log A \sum_{h=1}^{H} \sum_{(x_h, a_h) \in \mathcal{X}_h \times \mathcal{A}} \frac{\mu^\dagger_{1:h}(x_h, a_h)}{\eta} \\
&\leq \frac{X \log A}{\eta}.
\end{aligned}
$$

$\square$

## D.2 BOUNDING THE STABILITY TERM

To begin with, we first introduce the following lemma, which is a generalized version of Lemma D.7 by Bai et al. (2022). Intuitively, this lemma states that the one-step stability term can be bounded by the inner product between $\hat{\mu}$ and $\hat{\ell}^t$ as well as the summation of log-partition function $\log Z_1^t$.

**Lemma D.2.** *For given $\eta$ and any $\mu \in \Pi_{\max}$, we have*

$$D_\eta(\mu \| \hat{\mu}^{t+1}) - D_\eta(\mu \| \hat{\mu}^t) = \left\langle \mu, \hat{\ell}^t \right\rangle + \sum_{x_1 \in \mathcal{X}_1} \frac{1}{\eta_1(x_1)} \log Z_1^t(x_1). \quad (18)$$

*Proof.*

$$
\begin{aligned}
&D_\eta(\mu \| \hat{\mu}^{t+1}) - D_\eta(\mu \| \hat{\mu}^t) \\
&= \sum_{h=1}^{H} \sum_{(x_h, a_h) \in \mathcal{X}_h \times \mathcal{A}} \frac{\mu_{1:h}(x_h, a_h)}{\eta_h(x_h)} \log \frac{\hat{\mu}^t_h(a_h | x_h)}{\hat{\mu}^{t+1}_h(a_h | x_h)} \\
&= \sum_{h=1}^{H} \sum_{(x_h, a_h) \in \mathcal{X}_h \times \mathcal{A}} \frac{\mu^t_{1:h}(x_h, a_h)}{\eta_h(x_h)} \left( \eta_h(x_h) \hat{\ell}^t_h(x_h, a_h) - \sum_{x_{h+1} \in C(x_h, a_h)} \frac{\eta_h(x_h)}{\eta_{h+1}(x_{h+1})} \log Z^t_{h+1}(x_{h+1}) \right) \\
&\quad + \sum_{h=1}^{H} \sum_{x_h \in \mathcal{X}_h} \frac{\mu_{1:h-1}(x_h)}{\eta_h(x_h)} \log Z^t_h(x_h) \\
&= \left\langle \mu, \hat{\ell}^t \right\rangle + \sum_{h=1}^{H} \left[ -\sum_{x_{h+1} \in \mathcal{X}_{h+1}} \frac{\mu_{1:h}(x_{h+1})}{\eta_{h+1}(x_{h+1})} \log Z^t_{h+1}(x_{h+1}) + \sum_{x_h \in \mathcal{X}_h} \frac{\mu_{1:h-1}(x_h)}{\eta_h(x_h)} \log Z^t_h(x_h) \right] \\
&= \left\langle \mu, \hat{\ell}^t \right\rangle + \sum_{x_1 \in \mathcal{X}_1} \frac{1}{\eta_1(x_1)} \log Z^t_1(x_1).
\end{aligned}
$$

$\square$

From Lemma D.2 and setting $\eta_h(x_h) \equiv \eta$, we have

$$
\begin{aligned}
D_\eta(\hat{\mu}^t \| \hat{\mu}^{t+1}) &= \left\langle \hat{\mu}^t, \hat{\ell}^t \right\rangle + \sum_{x_1 \in \mathcal{X}_1} \frac{1}{\eta_1(x_1)} \log Z_1^t(x_1) \\
&= \left\langle \hat{\mu}^t, \hat{\ell}^t \right\rangle + \frac{1}{\eta} \sum_{x_1 \in \mathcal{X}_1} \log Z_1^t(x_1) .
\end{aligned} \tag{19}
$$

Hence, to bound the STABILITY term, it suffices to bound the log-partition function $\log Z_1^t$. To this end, roughly speaking, we prove that the summation of all $\log Z_1^t(x_1)$ for $x_1 \in \mathcal{X}_1$ can be bounded by the product between (a) the value of all the reachable $(x_h, a_h)$ in the element-wise product of the random vectors independently sampled from the categorical distributions specified by $\hat{\mu}^t(\cdot|x_h)$; and (b) the loss estimate at $(x_h, a_h)$. Compared with the analysis tailored to the importance-weighted loss estimate in previous works (Kozuno et al., 2021; Fiegel et al., 2023), where bounding similar log-partition function $\log Z_1^t(x_1)$ is easier and can be done by only considering the random variables sampled from the Bernoulli distributions along the experienced trajectory, our analysis for least-squares loss estimate is more challenging and also generalizes it in previous works.

### D.2.1 BOUNDING THE LOG-PARTITION FUNCTION $\log Z_h^t$

We first define $\boldsymbol{z}^t(x_h, \cdot) \in \{0, 1\}^A$, which is a random vector independently sampled from the categorical distribution parameterized by $\hat{\mu}_h^t(\cdot|x_h)$, by

$$
\boldsymbol{z}^t(x_h, \cdot) \sim \mathrm{Cat}(\hat{\mu}_h^t(\cdot|x_h)) ,
$$

such that $\mathbb{P}\left(\boldsymbol{z}^t(x_h, a_h) = 1\right) = \hat{\mu}_h^t(a_h|x_h)$.

Notice that

$$
\mathbb{E}\left[ \prod_{h'=1}^{h} \boldsymbol{z}^t(x_{h'}, a_{h'}) \right] = \hat{\mu}_{1:h}^t(x_h, a_h) .
$$

We also let

$$
\boldsymbol{z}^t{}_{h:h'}(x_{h'}, a_{h'}) = \prod_{h''=h}^{h'} \boldsymbol{z}^t(x_{h''}, a_{h''}) ,
$$

where $\{(x_{h''}, a_{h''})\}_{h'' \in [h, h']}$ is the unique path from $(x_h, a_h)$ to $(x_{h'}, a_{h'})$ (under perfect recall condition). Besides, we denote the product of $Z_{h+1}^t(x_{h+1})$ for all $x_{h+1} \in C(x_h, a_h)$ as

$$
\Xi^t(x_h, a_h) = \prod_{x_{h+1} \in C(x_h, a_h)} Z_{h+1}^t(x_{h+1}) ,
$$

so that

$$
\frac{1}{\eta} \sum_{x_1 \in \mathcal{X}_1} \log Z_1^t(x_1) = \frac{1}{\eta} \log \Xi^t(\emptyset) .
$$

Then, the following lemma shows that, $\Xi_h^t(x_h, a_h)$ is equivalent to the expectation of the exponentiation of the summation of $\boldsymbol{z}^t{}_{h+1:h'}(x_{h'}, a_{h'})\hat{\ell}_{h'}^t(x_{h'}, a_{h'})$, where $(x_{h'}, a_{h'})$ are all the reachable infoset-action pairs from $(x_h, a_h)$.

**Lemma D.3.** *For any $(x_h, a_h) \in \mathcal{X}_h \times \mathcal{A}$ and $h \in [H-1]$, we have*

$$
\Xi_h^t(x_h, a_h) = \mathbb{E}_{\boldsymbol{z}^t}\left[ \exp\left( -\eta \sum_{h'=h+1}^{H} \sum_{(x_{h'}, a_{h'}) \in C_{h'}(x_h, a_h)} \boldsymbol{z}^t{}_{h+1:h'}(x_{h'}, a_{h'})\hat{\ell}_{h'}^t(x_{h'}, a_{h'}) \right) \right] . \tag{20}
$$

As an immediate corollary of Lemma D.3, we have

$$
\Xi^t(\emptyset) = \mathbb{E}_{\boldsymbol{z}^t}\left[ \exp\left( -\eta \sum_{(x_h, a_h) \in \mathcal{X}_h \times \mathcal{A}} \boldsymbol{z}^t{}_{1:h}(x_h, a_h)\hat{\ell}_h^t(x_h, a_h) \right) \right] .
$$

*Proof.* We prove this by backward induction. For $h = H - 1$, we have

$$\Xi^t_{H-1}(x_{H-1}, a_{H-1}) = \prod_{x_H \in C(x_{H-1}, a_{H-1})} \sum_{a_H \in \mathcal{A}} \hat{\mu}^t_H(a_H | x_H) \exp(-\eta \hat{\ell}^t_H(x_H, a_H))$$

$$= \mathbb{E}_{\boldsymbol{z}^t} \left[ \exp \left( -\eta \boldsymbol{z}^t_{H:H}(x_H, a_H) \hat{\ell}^t_H(x_H, a_H) \right) \right].$$

Suppose Eq. (20) holds from $h' = h$ to $H$. Then for $h' = h - 1$, one can deduce that

$$\Xi^t_{h-1}(x_{h-1}, a_{h-1})$$

$$= \prod_{x_h \in C(x_{h-1}, a_{h-1})} \sum_{a_h \in \mathcal{A}} \hat{\mu}^t_h(a_h | x_h) \exp \left( -\eta \hat{\ell}^t_h(x_h, a_h) \right) \Xi^t_h(x_h, a_h)$$

$$= \prod_{x_h \in C(x_{h-1}, a_{h-1})} \sum_{a_h \in \mathcal{A}} \hat{\mu}^t_h(a_h | x_h) \exp \left( -\eta \hat{\ell}^t_h(x_h, a_h) \right)$$

$$\cdot \mathbb{E}_{\boldsymbol{z}^t} \left[ \exp \left( -\eta \sum_{h'=h+1}^{H} \sum_{(x_{h'}, a_{h'}) \in C_{h'}(x_h, a_h)} \boldsymbol{z}^t_{h+1:h'}(x_{h'}, a_{h'}) \hat{\ell}^t_{h'}(x_{h'}, a_{h'}) \right) \right]$$

$$= \prod_{x_h \in C(x_{h-1}, a_{h-1})} \mathbb{E}_{\boldsymbol{z}^t, a_h} \left[ \exp \left( -\eta \sum_{h'=h+1}^{H} \boldsymbol{z}^t(x_h, a_h) \right. \right.$$

$$\left. \left. \cdot \left( \sum_{x_{h'}, a_{h'} \in C_{h'}(x_h, a_h)} \boldsymbol{z}^t_{h+1:h'}(x_{h'}, a_{h'}) \hat{\ell}^t_{h'}(x_{h'}, a_{h'}) + \hat{\ell}^t_h(x_h, a_h) \right) \right) \right]$$

$$= \mathbb{E}_{\boldsymbol{z}^t} \left[ \exp \left( -\eta \sum_{h'=h}^{H} \sum_{x_{h'}, a_{h'} \in C_{h'}(x_{h-1}, a_{h-1})} \boldsymbol{z}^t_{h:h'}(x_{h'}, a_{h'}) \hat{\ell}^t_{h'}(x_{h'}, a_{h'}) \right) \right],$$

which completes the proof. $\square$

### D.2.2 BOUNDING THE VARIANCE OF THE LOSS ESTIMATE

The following lemma bounds the variance of the loss estimate.

**Lemma D.4.** *For and $h \in [H]$ and any $(x_h, a_h) \in \mathcal{X}_h \times \mathcal{A}$, it holds that $|\hat{\ell}^t_h(x_h, a_h)| \leq \frac{1}{\gamma \rho}$.*

*Proof.* First notice that for any $\nu^t$ and any $(x_h, a_h) \in \mathcal{X}_h \times \mathcal{A}$, we have

$$\left\| \phi^{\nu^t}(x_h, a_h) \right\|_2$$

$$= \left\| -\sum_{(s_h, b_h) \in x_h \times \mathcal{B}} p_{1:h}(s_h) \nu^t_{1:h}(y(s_h), b_h) \phi(s_h, a_h, b_h) \right\|_2$$

$$\leq \sum_{(s_h, b_h) \in x_h \times \mathcal{B}} p_{1:h}(s_h) \nu^t_{1:h}(y(s_h), b_h) \left\| \phi(s_h, a_h, b_h) \right\|_2$$

$$\overset{(i)}{\leq} \sum_{(s_h, b_h) \in x_h \times \mathcal{B}} p_{1:h}(s_h) \nu^t_{1:h}(y(s_h), b_h)$$

$$\overset{(ii)}{\leq} 1, \tag{21}$$

where $(i)$ is due to Assumption 2.1; and $(ii)$ follows from the proof of Lemma 2 by Kozuno et al. (2021).

Recall that $\mu^t = (1 - \gamma)\hat{\mu}^t + \gamma\pi$. Let $\Phi_h^t := \left\{ \phi^{\nu^t}(x_h, a_h) \right\}_{(x_h, a_h) \in \mathcal{X}_h \times \mathcal{A}}$. It is then clear that

$$
\begin{aligned}
|\hat{\ell}_h^t(x_h, a_h)| &= |\phi^{\nu^t}(x_h, a_h)^\top \boldsymbol{Q}_{\mu^t,h}^{-1} \phi_t r_h(s_h, a_h, b_h)| \\
&\overset{(i)}{\leq} |\phi^{\nu^t}(x_h, a_h)^\top \boldsymbol{Q}_{\mu^t,h}^{-1} \phi_t| \\
&\overset{(ii)}{\leq} \|\phi^{\nu^t}(x_h, a_h)\|_{\boldsymbol{Q}_{\mu^t,h}^{-1}} \cdot \sup_{\phi \in \Phi_h} \|\phi\|_{\boldsymbol{Q}_{\mu^t,h}^{-1}} \\
&\leq \sup_{\phi \in \Phi_h^t} \|\phi\|_{\boldsymbol{Q}_{\mu^t,h}^{-1}}^2 \\
&\leq \sup_{\phi \in \Phi_h^t} \|\phi\|_{(\gamma \boldsymbol{Q}_{\pi^t,h})^{-1}}^2 \\
&\leq \sup_{\phi \in \Phi_h^t} \|\phi\|_{(\gamma\rho\boldsymbol{I})^{-1}}^2 \\
&\overset{(iii)}{\leq} \frac{1}{\gamma\rho},
\end{aligned}
$$

where $(i)$ is because $|r_h(s_h, a_h, b_h)| \leq 1$; $(ii)$ is by the Cauchy-Schwarz inequality; and $(iii)$ comes from Eq. (21). $\qquad\square$

### D.2.3 Final Proof the Stability Term

We are now ready to bound the Stability term.

**Lemma D.5.** *The* Stability *term is bounded by*

$$
\text{Stability} \leq \eta T X H d.
$$

*Proof.* Plugging Eq. (20) into Eq. (19), we have

$$\left\langle \hat{\mu}^t, \hat{\ell}^t \right\rangle + \frac{1}{\eta} \sum_{x_1 \in X_1} \log Z_1^t(x_1)$$

$$= \left\langle \hat{\mu}^t, \hat{\ell}^t \right\rangle + \frac{1}{\eta} \log \Xi^t(\emptyset)$$

$$= \left\langle \hat{\mu}^t, \hat{\ell}^t \right\rangle + \frac{1}{\eta} \log \mathbb{E}_{\boldsymbol{z}^t} \left[ \exp \left( -\eta \underbrace{\sum_{(x_h, a_h) \in \mathcal{X}_h \times \mathcal{A}} \boldsymbol{z}^t_{1:h}(x_h, a_h) \hat{\ell}_h^t(x_h, a_h)}_{\spadesuit} \right) \right]$$

$$\overset{(i)}{\leq} \left\langle \hat{\mu}^t, \hat{\ell}^t \right\rangle + \frac{1}{\eta} \log \mathbb{E}_{\boldsymbol{z}^t} \left[ 1 - \eta \sum_{(x_h, a_h) \in \mathcal{X}_h \times \mathcal{A}} \boldsymbol{z}^t_{1:h}(x_h, a_h) \hat{\ell}_h^t(x_h, a_h) + \left( \eta \sum_{(x_h, a_h) \in \mathcal{X}_h \times \mathcal{A}} \boldsymbol{z}^t_{1:h}(x_h, a_h) \hat{\ell}_h^t(x_h, a_h) \right)^2 \right]$$

$$\overset{(ii)}{\leq} \left\langle \hat{\mu}^t, \hat{\ell}^t \right\rangle - \frac{1}{\eta} \mathbb{E}_{\boldsymbol{z}^t} \left[ \eta \sum_{(x_h, a_h) \in \mathcal{X}_h \times \mathcal{A}} \boldsymbol{z}^t_{1:h}(x_h, a_h) \hat{\ell}_h^t(x_h, a_h) \right]$$

$$+ \frac{1}{\eta} \mathbb{E}_{\boldsymbol{z}^t} \left[ \left( \eta \sum_{(x_h, a_h) \in \mathcal{X}_h \times \mathcal{A}} \boldsymbol{z}^t_{1:h}(x_h, a_h) \hat{\ell}_h^t(x_h, a_h) \right)^2 \right]$$

$$= \left\langle \hat{\mu}^t, \hat{\ell}^t \right\rangle - \sum_{(x_h, a_h) \in \mathcal{X}_h \times \mathcal{A}} \mathbb{E}_{\boldsymbol{z}^t} \left[ \boldsymbol{z}^t_{1:h}(x_h, a_h) \hat{\ell}_h^t(x_h, a_h) \right]$$

$$+ \frac{1}{\eta} \mathbb{E}_{\boldsymbol{z}^t} \left[ \left( \eta \sum_{(x_h, a_h) \in \mathcal{X}_h \times \mathcal{A}} \boldsymbol{z}^t_{1:h}(x_h, a_h) \hat{\ell}_h^t(x_h, a_h) \right)^2 \right]$$

$$= \frac{1}{\eta} \mathbb{E}_{\boldsymbol{z}^t} \left[ \left( \eta \sum_{(x_h, a_h) \in \mathcal{X}_h \times \mathcal{A}} \boldsymbol{z}^t_{1:h}(x_h, a_h) \hat{\ell}_h^t(x_h, a_h) \right)^2 \right]$$

$$\overset{(iii)}{\leq} \eta \left( \sum_{(x_h, a_h) \in \mathcal{X}_h \times \mathcal{A}} \hat{\mu}^t_{1:h}(x_h, a_h) \right) \left( \sum_{(x_h, a_h) \in \mathcal{X}_h \times \mathcal{A}} \hat{\mu}^t_{1:h}(x_h, a_h) \hat{\ell}_h^t(x_h, a_h)^2 \right)$$

$$\leq \eta X \left( \sum_{(x_h, a_h) \in \mathcal{X}_h \times \mathcal{A}} \hat{\mu}^t_{1:h}(x_h, a_h) \hat{\ell}_h^t(x_h, a_h)^2 \right), \tag{22}$$

where $|\spadesuit| \leq 1$ follows from setting $\gamma \geq \eta X \rho^{-1}$ and Lemma D.4 in conjunction with Assumption 3.2; $(i)$ is from $\exp(-x) \leq 1 - x + x^2$ for $x \geq -1$; $(ii)$ comes from $\forall x > 0, \log x \leq x - 1$; $(iii)$ is by the Cauchy–Schwarz inequality.

The proof is then concluded by taking summation of Eq. (22) over $t$ and using Lemma B.1.

$\square$

## E    PROOF OF REGRET GUARANTEES OF LSFTRL

To start with, notice that $\Pi_{\max}$ is an affine subspace of $\mathbb{R}_{\geq 0}^{XA}$ satisfying $X$ linear constraints: for any $x_h \in \mathcal{X}$,

$$\sum_{a_h \in \mathcal{A}} \mu_{1:h}(x_h, a_h) = \mu_{1:h-1}(x_{h-1}, a_{h-1}),$$

where $(x_{h-1}, a_{h-1})$ is the unique predecessor of $x_h$ under perfect recall condition. Thus $\Pi_{\max}$ can be decomposed as $\Pi_{\max} = (F + u) \cap \mathbb{R}_{\geq 0}^{XA}$ where $F$ is a linear subspace and $u \in \Pi_{\max}$.

With slight abuse of notations, we further denote $\Psi(\mu) = \frac{1}{\eta} \sum_{h=1}^{H} \Psi_h \left( p_{1:h}^\star \cdot \mu_{1:h} \right)$ and define its convex conjugate function $\Psi^\star$ on $\mathbb{R}_{\geq 0}^{XA}$:

$$\Psi^\star(\boldsymbol{y}) := \sup_{\boldsymbol{x} \in \mathbb{R}_{\geq 0}^{AX}} \langle \boldsymbol{x}, \boldsymbol{y} \rangle - \Psi(\boldsymbol{x}). \tag{23}$$

Also, we denote $D_{\Psi^*}(\boldsymbol{x}, \boldsymbol{y}) = \Psi^\star(\boldsymbol{x}) - \Psi^\star(\boldsymbol{y}) - \langle \nabla \Psi^\star(\boldsymbol{y}), \boldsymbol{x} - \boldsymbol{y} \rangle$ as the Bregman divergence induced by $\Psi^\star$. The following lemma shows the canonical regret decomposition of FTRL algorithm ([Zimmert & Seldin](), 2019; [Lattimore & Szepesvári](), 2020).

**Lemma E.1.** *The regret of* `LSFTRL` *can be decomposed as*

$$\mathfrak{R}_{\max}^T \leq \underbrace{\max_{\mu \in \Pi_{\max}} \left[ -\Psi(\mu) \right]}_{\text{PENALTY}} + \underbrace{\mathbb{E} \left[ \sum_{t=1}^T D_{\Psi^*}(\nabla \Psi(\mu^t) - \hat{\ell}^t, \nabla \Psi(\mu^t)) \right]}_{\text{STABILITY}}.$$

*Proof.* Let $\mu^\dagger \in \Pi_{\max}$ be some realization plan. For all $t \in [T]$, the instantaneous regret against $\mu^\dagger$ at step $t$ can be decomposed into

$$\left\langle \mu^t - \mu^\dagger, \hat{\ell}^t \right\rangle = \left[ \Phi \left( -\hat{L}^{t-1} \right) - \Phi \left( -\hat{L}^t \right) - \left\langle \mu^\dagger, \hat{\ell}^t \right\rangle \right] + \left[ \left\langle \mu^t, \hat{\ell}^t \right\rangle + \Phi \left( -\hat{L}^t \right) - \Phi \left( -\hat{L}^{t-1} \right) \right],$$

where $\Phi(\boldsymbol{y}) := \sup_{\boldsymbol{\mu} \in \Pi_{\max}} \langle \boldsymbol{\mu}, \boldsymbol{y} \rangle - \Psi(\boldsymbol{\mu})$.

Taking summation of the above display over $t$ yields

$$\sum_{t=1}^T \left[ \Phi \left( -\hat{L}^{t-1} \right) - \Phi \left( -\hat{L}^t \right) - \left\langle \mu^\dagger, \hat{\ell}^t \right\rangle \right]$$

$$= \Phi(0) - \Phi \left( -\hat{L}^t \right) - \left\langle \mu^\dagger, \hat{L}^t \right\rangle$$

$$\overset{(i)}{\leq} \max_{\mu \in \Pi_{\max}} \left[ -\Psi(\mu) \right] + \Psi \left( \mu^\dagger \right)$$

$$\overset{(ii)}{\leq} \max_{\mu \in \Pi_{\max}} \left[ -\Psi(\mu) \right],$$

where $(i)$ comes from $\mu^\dagger \in \Pi_{\max}$; and $(ii)$ is due to the fact that $\Psi$ is a non-positive function.

On the other hand, due to that $\Pi_{\max} = (F + u) \cap \mathbb{R}_{\geq 0}^{XA}$, we have

$$\left\langle \mu^t, \hat{\ell}^t \right\rangle + \Phi \left( -\hat{L}^t \right) - \Phi \left( -\hat{L}^{t-1} \right)$$

$$\overset{(i)}{=} \left\langle \mu^t, \hat{\ell}^t \right\rangle + \Phi \left( \nabla \Psi \left( \mu^t \right) + \boldsymbol{g}^t - \hat{\ell}^t \right) - \Phi \left( \nabla \Psi \left( \mu^t \right) + \boldsymbol{g}^t \right)$$

$$\overset{(ii)}{=} \left\langle \mu^t, \hat{\ell}_t \right\rangle + \Phi \left( \nabla \Psi \left( \mu^t \right) - \hat{\ell}^t \right) - \Phi \left( \nabla \Psi \left( \mu^t \right) \right)$$

$$\overset{(iii)}{\leq} \left\langle \mu^t, \hat{\ell}_t \right\rangle + \Psi^* \left( \nabla \Psi \left( \mu^t \right) - \hat{\ell}^t \right) - \Psi^* \left( \nabla \Psi \left( \mu^t \right) \right)$$

$$\overset{(iv)}{=} D_{\Psi^*} \left( \nabla \Psi \left( \mu^t \right) - \hat{\ell}^t, \nabla \Psi \left( \mu^t \right) \right),$$

where $(i)$ follows from $\hat{L}^{t-1} + \nabla\Psi(\mu^t) + g^t = 0$ for $g^t \in F^{\perp}$; $(ii)$ is due to the fact that $y \in \mathbb{R}^{XA}$,

$$
\begin{aligned}
\Phi(y + g^t) &= \sup_{\mu \in (F+u) \cap \mathbb{R}^{XA}_{\geq 0}} \langle \mu, y + g^t \rangle - \Psi(\mu) \\
&= \left( \sup_{\mu \in F \cap \mathbb{R}^{XA}_{\geq 0}} \langle \mu, y + g^t \rangle - \Psi(\mu) \right) + \langle u, y + g^t \rangle \\
&= \left( \sup_{\mu \in F \cap \mathbb{R}^{XA}_{\geq 0}} \langle \mu, y \rangle - \Psi(\mu) \right) + \langle u, y + g^t \rangle \\
&= \left( \sup_{\mu \in (F+u) \cap \mathbb{R}^{XA}_{\geq 0}} \langle \mu, y \rangle - \Psi(\mu) \right) + \langle u, g^t \rangle \\
&= \Phi(y) + \langle u, g^t \rangle \ ;
\end{aligned}
$$

$(iii)$ is by the observation that $\forall y \in \mathbb{R}^{XA}$, $\Phi(y) \leq \Psi^*(y)$ and $\mu^t = \arg\max_{x \in \mathbb{R}^{XA}_{\geq 0}} \langle x, \nabla\Psi(\mu^t) \rangle - \Psi(x)$ which implies that $\Phi(\nabla\Psi(\mu^t)) = \Psi^*(\nabla\Psi(\mu^t))$; and $(iv)$ comes from the definition of $D_{\Psi^*}(x, y)$.

$\square$

The following lemma shows that the STABILITY term can be bounded by the variance of the loss estimate, which is the expected version of Lemma E.6 in Fiegel et al. (2023). We also present the proof here for completeness.

**Lemma E.2.** *Let $v_t = D_{\Psi^*}(\nabla\Psi(\mu^t) - \hat{\ell}^t, \nabla\Psi(\mu^t))$ for all $t \in [T]$. Then, it holds that*

$$
\text{STABILITY} = \mathbb{E}\left[ \sum_{t=1}^{T} v_t \right].
$$

*Furthermore, we have*

$$
\mathbb{E}\left[ \sum_{t=1}^{T} v_t \right] \leq \mathbb{E}\left[ \frac{\eta}{2} \sum_{t=1}^{T} \sum_{h=1}^{H} \sum_{(x_h, a_h) \in \mathcal{X}_h \times \mathcal{A}} \frac{1}{p^*_{1:h}(x_h)} \mathbb{E}^{\mu^t, \nu^t}\left[ \mu^t_{1:h}(x_h, a_h) \hat{\ell}^t(x_h, a_h)^2 \right] \right].
$$

*Proof.* To begin with, for all $t \in [T]$, we define

$$
f_t(u) = D_{\Psi^*}\left( \nabla\Psi(\mu^t) - u\hat{\ell}^t, \nabla\Psi(\mu^t) \right),
$$

for $u \in [0, 1]$, such that $f_t(0) = 0$ and $f_t(1) = v_t$. Also notice that $\text{dom}(\Psi^*) = \mathbb{R}^{XA}_{\geq 0}$ and both $\Psi$ and $\Psi^*$ can be decomposed according to each infoset-action pair $(x_h, a_h)$. Specifically, we have

$$
\Psi(\mu) = \sum_{h=1}^{H} \sum_{(x_h, a_h) \in \mathcal{X}_h \times \mathcal{A}} \Psi_{x_h, a_h}(\mu_{1:h}(x_h, a_h)),
$$

and

$$
\Psi^*(y) = \sum_{h=1}^{H} \sum_{(x_h, a_h) \in \mathcal{X}_h \times \mathcal{A}} \Psi^*_{x_h, a_h}(y(x_h, a_h)).
$$

Then the derivative of $f_t$ can be expressed as

$$
f'_t(u) = \sum_{h=1}^{H} \sum_{(x_h, a_h) \in \mathcal{X}_h \times \mathcal{A}} \hat{\ell}^t_h(x_h, a_h)\left[ \mu^t_{1:h}(x_h, a_h) - \nabla\Psi^*_{x_h, a_h}\left( \nabla\Psi_{x_h, a_h}(\mu^t_{1:h}(x_h, a_h)) - u\hat{\ell}^t_h(x_h, a_h) \right) \right].
$$

(24)

Moreover, recall that we choose negative entropy as the potential function. Therefore, it holds that

$$\nabla \Psi_{x_h, a_h} \left( \mu_{1:h} \left( x_h, a_h \right) \right) = \frac{p^\star_{1:h} \left( x_h \right)}{\eta} \left[ \log \left( p^\star_{1:h} \left( x_h \right) \mu_{1:h} \left( x_h, a_h \right) \right) + 1 \right],$$

$$\nabla \Psi^\star_{x_h, a_h} \left( y \left( x_h, a_h \right) \right) = \exp \left[ \frac{\eta}{p^\star_{1:h} \left( x_h \right)} \left( y \left( x_h, a_h \right) \right) - 1 - \log \left( p^\star_{1:h} \left( x_h \right) \right) \right],$$

and

$$\nabla \Psi^\star_{x_h, a_h} \left( \nabla \Psi_{x_h, a_h} \left( \mu^t_{1:h} \left( x_h, a_h \right) \right) - u \hat{\ell}^t_h \left( x_h, a_h \right) \right)$$

$$= \exp \left[ \frac{\eta}{p^\star_{1:h} \left( x_h \right)} \left( \frac{p^\star_{1:h} \left( x_h \right)}{\eta} \log \left( p^\star_{1:h} \left( x_h \right) \mu^t_{1:h} \left( x_h, a_h \right) \right) - u \hat{\ell}^t_h \left( x_h, a_h \right) \right) - \log \left( p^\star_{1:h} \left( x_h \right) \right) \right]$$

$$= \mu^t_{1:h} \left( x_h, a_h \right) \exp \left[ -u \frac{\eta \hat{\ell}^t_h \left( x_h, a_h \right)}{p^\star_{1:h} \left( x_h \right)} \right]$$

$$\geq \mu^t_{1:h} \left( x_h, a_h \right) \left[ 1 - u \frac{\eta \hat{\ell}^t_h \left( x_h, a_h \right)}{p^\star_{1:h} \left( x_h \right)} \right], \tag{25}$$

where the last inequality follows from $e^{-x} \geq 1 - x$ for all $x \in \mathbb{R}$.

Substituting Eq. (25) into Eq. (24) shows that

$$f'_t(u) \leq u \sum_{h=1}^{H} \sum_{(x_h, a_h) \in \mathcal{X}_h \times \mathcal{A}} \hat{\ell}^t_h \left( x_h, a_h \right) \mu^t_{1:h} \left( x_h, a_h \right) \frac{\eta \hat{\ell}^t_h \left( x_h, a_h \right)}{p^\star_{1:h} \left( x_h \right)}.$$

The proof is concluded by integrating the above display from $0$ to $1$ over $u$ and taking the expectation on both sides. $\qquad \square$

### E.1    Additional Discussions on Assumption 4.1

Intuitively, this assumption says that the environment transition $\mathbb{P}$ and opponent's policy $\nu_t$ are *balanced* enough in the sense that $p^{\nu^t}_{1:h} \left( x_h \right)$ induced by $\mathbb{P}$ and $\nu^t$ is not too large compared with the "balanced" transition $p^\star_{1:h}(x_2)$ for any $x_1, x_2 \in \mathcal{X}_h$ and $h \in [H]$. Indeed, consider the case where the game tree is an $k$-ary tree and $\mathbb{P}$ is uniform distribution at every underlying state $s$, then it holds that $\lambda = 1$. On the other hand, the worst-case scenario is that $\lambda = \mathcal{O}(X)$ if $p^{\nu^t}_{1:H}(x_1) = 1$ for some $x_1 \in \mathcal{X}_H$. Nevertheless, this extreme case is very unlikely to happen in practice unless it simultaneously happens that (a) the environment state transitions along the trajectory $\{(s_h, a_h, b_h)\}_{h \in [H-1]}$ leading to $s_H$ s.t. $x(s_H) = x_1$ satisfy $p_h(s_{h+1} \mid s_h, a_h, b_h) = 1$ for all $(s_h, a_h, b_h)$ along the trajectory; and (b) the opponent *knows* the underlying environment transitions and the mapping $y : \mathcal{S} \to \mathcal{Y}$ so that the opponent can ensure $\nu^t_{1:H-1} \left( y \left( s_{H-1} \right), b_{H-1} \right) = 1$ by setting $\nu^t(b_h \mid y(s_h)) = 1$ for all $(s_h, b_h)$ along the trajectory.

### E.2    Proof of Theorem 4.2

In this section, we provide the proof of Theorem 4.2, which takes $p^\star$ as the transition probability function over infoset-action pairs.

*Proof of Theorem 4.2.* Combining Lemma E.1, E.3 and E.4, with $p^\star$ computed by `Computing-`$p^\star$, we have that

$$\mathfrak{R}^T_{\max} \leq \text{Penalty} + \text{Stability}$$

$$\leq \frac{H}{\eta} \log \left( XA \right) + \frac{\eta}{2} TH d\lambda, \tag{26}$$

which along with choosing $\eta = \sqrt{\frac{2 \log(AX)}{Td\lambda}}$ finishes the proof.

$\qquad \square$

We note that leveraging $\beta$ as well as Assumption 4.1 necessitates identifying a transition probability function $p^\star$ with its minimum visitation probability achieving $\beta$. Finding such $p^\star$ is done by the procedure illustrated in Appendix E.2.3.

### E.2.1 BOUNDING THE PENALTY TERM

The lemma below directly follows from Lemma E.5 of Fiegel et al. (2023), with its proof provided here for completeness.

**Lemma E.3.** *For any fixed learning rate $\eta$ and transition probability function $p^\star$, it holds that*

$$\text{PENALTY} \leq \frac{H}{\eta} \log{(XA)}\ .$$

*Proof.* It is clear that

$$-\Psi(\mu) = -\frac{1}{\eta} \sum_{h=1}^{H} \Psi_h \left( p_{1:h}^\star \cdot \mu_{1:h} \right) \overset{(i)}{\leq} \frac{1}{\eta} \sum_{h=1}^{H} \log{(X_h A)} \leq \frac{1}{\eta} \sum_{h=1}^{H} \log{(XA)} = \frac{H}{\eta} \log{(XA)}\ ,$$

where $(i)$ comes from Lemma A.1. $\qquad\square$

### E.2.2 BOUNDING THE STABILITY TERM

**Lemma E.4.** *For any fixed learning rate $\eta$ and transition probability function $p^\star$, it holds that*

$$\text{STABILITY} \leq \frac{\eta}{2} THd\lambda\ .$$

*Proof.* Recall that $\beta = \max_{\tilde{p} \in \mathbb{P}^\star} \min_{h \in [H], x_h \in \mathcal{X}_h} \tilde{p}_{1:h}(x_h)$. Then, one can see that

$$\text{STABILITY} \leq \mathbb{E}\left[ \frac{\eta}{2} \sum_{t=1}^{T} \sum_{h=1}^{H} \sum_{(x_h, a_h) \in \mathcal{X}_h \times \mathcal{A}} \frac{1}{p_{1:h}^\star(x_h)} \mathbb{E}^{\mu^t, \nu^t} \left[ \mu_{1:h}^t(x_h, a_h) \hat{\ell}^t(x_h, a_h)^2 \right] \right]$$

$$\leq \mathbb{E}\left[ \frac{\eta}{2} \sum_{t=1}^{T} \sum_{h=1}^{H} \frac{1}{\beta} \operatorname{tr}\left( \sum_{(x_h, a_h) \in \mathcal{X}_h \times \mathcal{A}} p^{\nu^t}(x_h) \mu_{1:h}^t(x_h, a_h) \phi^{\nu^t}(x_h, a_h) \phi^{\nu^t}(x_h, a_h)^\top \boldsymbol{Q}_{\mu^t, h}^{-1} \right) \right]$$

$$\overset{(i)}{\leq} \mathbb{E}\left[ \frac{\eta}{2} \sum_{t=1}^{T} \sum_{h=1}^{H} \lambda \operatorname{tr}\left( \sum_{(x_h, a_h) \in \mathcal{X}_h \times \mathcal{A}} \mu_{1:h}^t(x_h, a_h) \phi^{\nu^t}(x_h, a_h) \phi^{\nu^t}(x_h, a_h)^\top \boldsymbol{Q}_{\mu^t, h}^{-1} \right) \right]$$

$$\overset{(ii)}{\leq} \frac{\eta}{2} THd\lambda\ ,$$

where $(i)$ is due to Assumption 4.1; and $(ii)$ comes from Lemma B.1. $\qquad\square$

### E.2.3 COMPUTING $p^\star$

The procedure `Computing-`$p^\star$ can compute $p^\star$ in Eq. (9), via backward dynamic programming in $\mathcal{O}(XA)$ time.

### E.3 PROOF OF THEOREM 4.4

This section presents the proof of Theorem 4.4.

*Proof of Theorem 4.4.* Combining Lemma E.1, E.5 and E.6, with $p_{1:h}^\star(x_h) \equiv 1$ for all $x_h \in \mathcal{X}$, we have that

$$\mathfrak{R}_{\max}^T \leq \text{PENALTY} + \text{STABILITY}$$
$$\leq \frac{X(1 + \log A)}{\eta} + \frac{\eta}{2} THd\ , \tag{27}$$

which concludes the proof by noticing that $\eta = \sqrt{\frac{2X \log A}{THd}}$. $\qquad\square$

---

**Algorithm 5** `Computing-`$p^\star$

---

1: **Input:** Tree-like structure of $\mathcal{X} \times \mathcal{A}$.
2: **Initialization:** Transition array $p[\cdot]$ of size $X$; auxiliary array $f[\cdot]$ of size $X$, $C[\cdot, \cdot]$ of size $X \times A$. For all $x_H$ in $\mathcal{X}_H$, initialize $f[x_H] = 1$.
3: **for** $h = H - 1$ to $1$ **do**
4:     **for** $x_h$ in $\mathcal{X}_h$ **do**
5:         **for** $a_h$ in $\mathcal{A}$ **do**
6:             Compute $C[x_h, a_h] = \sum_{x_{h+1} \in C(x_h, a_h)} f[x_{h+1}]$,
7:             Compute $f[x_h] = \max_{a \in \mathcal{A}} C[x_h, a]$.
8:         **end for**
9:     **end for**
10: **end for**
11: **for** $x_1$ in $\mathcal{X}_1$ **do**
12:     Compute $p[x_1] = \frac{f[x_1]}{\sum_{x_1 \in \mathcal{X}_1} f[x_1]}$.
13: **end for**
14: **for** $h = 1$ to $H - 1$ **do**
15:     **for** $x_h, a_h$ in $\mathcal{X}_h \times \mathcal{A}$ **do**
16:         **for** $x_{h+1}$ in $C(x_h, a_h)$ **do**
17:             Compute $p[x_{h+1}] = p[x_h] \cdot \frac{f[x_{h+1}]}{\sum_{x_{h+1} \in C(x_h, a_h)} f[x_{h+1}]}$.
18:         **end for**
19:     **end for**
20: **end for**
21: **return** $p$.

---

### E.3.1 BOUNDING THE PENALTY TERM

To bound the PENALTY term, we establish a refined analysis, which shaves off an $\mathcal{O}(\sqrt{A})$ factor compared with the direct combination of Lemma E.5 of Fiegel et al. (2023) and the setting of $p^\star_{1:h}(x_h) \equiv 1$.

**Lemma E.5.** *Setting* $p^\star_{1:h}(x_h) \equiv 1$ *for all* $x_h \in \mathcal{X}$. *For any fixed learning rate* $\eta$, *it holds that*

$$\text{PENALTY} \leq \frac{X(1 + \log A)}{\eta}.$$

*Proof.*

$$-\Psi(\mu) = -\frac{1}{\eta}\sum_{h=1}^{H}\Psi_h\left(p^\star_{1:h}\cdot\mu_{1:h}\right)$$

$$= -\frac{1}{\eta}\sum_{h=1}^{H}\sum_{(x_h,a_h)\in\mathcal{X}_h\times\mathcal{A}}\mu_{1:h}(x_h,a_h)\log\mu_{1:h}(x_h,a_h)$$

$$= -\frac{1}{\eta}\sum_{h=1}^{H}\sum_{(x_h,a_h)\in\mathcal{X}_h\times\mathcal{A}}\mu_{1:h-1}(x_h)\mu_{1:h}(a_h|x_h)\left(\log\mu_{1:h-1}(x_h)+\log\mu_h(a_h|x_h)\right)$$

$$= -\frac{1}{\eta}\sum_{h=1}^{H}\left(\sum_{x_h\in\mathcal{X}_h}\mu_{1:h-1}(x_h)\left(\log\mu_{1:h-1}(x_h)+\sum_{a_h\in\mathcal{A}}\mu_h(a_h|x_h)\log\mu_h(a_h|x_h)\right)\right)$$

$$\leq \frac{1}{\eta}\sum_{h=1}^{H}\left(\sum_{x_h\in\mathcal{X}_h}-\mu_{1:h-1}(x_h)\log\mu_{1:h-1}(x_h)+\sum_{x_h\in\mathcal{X}_h}\mu_{1:h-1}(x_h)\log A\right)$$

$$\overset{(i)}{\leq} \frac{1}{\eta}\sum_{h=1}^{H}\left(X_h+X_h\log A\right)$$

$$= \frac{X(1+\log A)}{\eta}.$$

Here $(i)$ comes from the fact that $-x\log x\leq 1$ for all $x\in[0,1]$. $\qquad\square$

### E.3.2 BOUNDING THE STABILITY TERM

The upper bound of STABILITY term when setting $p^\star_{1:h}(x_h)\equiv 1$ is guaranteed in the following lemma, the proof of which is omitted since it is essentially the same as that of Lemma E.4.

**Lemma E.6.** *Setting $p^\star_{1:h}(x_h)\equiv 1$ for all $x_h\in\mathcal{X}$. For any fixed learning rate $\eta$, it holds that*

$$\text{STABILITY} \leq \frac{\eta}{2}\mathbb{E}\left[\sum_{t=1}^{T}\sum_{h=1}^{H}\sum_{(x_h,a_h)\in\mathcal{X}_h\times\mathcal{A}}\mathbb{E}^{\mu^t,\nu^t}\left[\mu^t_{1:h}(x_h,a_h)\hat{\ell}^t(x_h,a_h)^2\right]\right]$$

$$\leq \frac{\eta}{2}THd.$$

## F PROOF OF REGRET LOWER BOUND

In this section, we present the proof of Theorem 4.6.

*Proof of Theorem 4.6.* We consider an $A$-ary tree POMG instance, in which

- $B=1$ so that there is actually no opponent effectively (and hence the dependence on the opponent's action $b$ is omitted in what follows);

- $X_h = S_h = A^{h-1}$ for all $h\in[H]$, which means that $\mathcal{X}_h = \mathcal{S}_h$ and there is actually no partial observability;

- $r_h(s,a)=0$ for all $h\in[H-1]$, and $r_H(s,a)$ is a reward sampled from Bernoulli distribution $\text{Ber}(\bar{r}_H(s,a))$ with mean $\bar{r}_H(s,a) = \langle\phi(s,a),\theta\rangle$.

By the construction, there exists a unique action sequence $(a_1,\ldots,a_{h-1})$ that determines $s_h$ (and hence $x_h$) and the transition is deterministic. Following similar arguments by Bai et al. (2022); Fiegel et al. (2023), it can be shown that if algorithm Alg achieves regret $\mathfrak{R}^T_{\max}$ on this POMG instance, then Alg can be used to tackle a stochastic linear bandit problem with $A^H$ "arms" and

obtain the regret with the same order as $\mathfrak{R}_{\max}^T$, where the reward for "arm" $(a_1, a_2, \ldots, a_H)$ (*i.e.*, $(s_H, a_H)$) is sampled from $\text{Ber}(\langle \phi(s_H, a_H), \boldsymbol{\theta} \rangle)$.

We now first consider the case when $H \geq d$. In this case, $\phi$ and $\boldsymbol{\theta}$ satisfy $\phi(s, a)_{[1:d-1]} \in \{-1, 1\}^{d-1}$, $\phi(s, a)_d = 1/4$, $\boldsymbol{\theta}_{[1:d-1]} \in \{-\Delta, \Delta\}^{d-1}$ with $\Delta = 1/(8\sqrt{2T})$ and $\boldsymbol{\theta}_d = 1$. Moreover, since $|\mathcal{S}_H \times \mathcal{A}| = A^{H-1} \cdot A = A^H$ as well as $H \geq d$ and $A \geq 2$, $\phi$ can be chosen such that $\{\phi(s, a)_{[1:d-1]}\}_{(s,a) \in \mathcal{S}_H \times \mathcal{A}} = \{-1, 1\}^{d-1}$ (omitting the duplicate feature vectors). Then by canonical analysis for the regret lower bound of stochastic linear bandits (see, *e.g.*, Theorem 24.1 by Lattimore & Szepesvári (2020); Lemma 25 by Zhou et al. (2021)), there exists a $\boldsymbol{\theta}_{[1:d-1]}^{\text{Alg}} \in \{-\Delta, \Delta\}^{d-1}$ such that $\mathfrak{R}^T \geq (d-1)\sqrt{T}/(16\sqrt{2}) = \Omega(\sqrt{d^2 T})$.

In case when $H < d$, we can choose $\phi$ such that the stochastic linear bandit problem, on which Alg suffers the same regret as on the POMG instance, has $2^H$ distinct feature vectors since $A \geq 2$ and $A^H \geq 2^H$. Then by similar reasoning of the construction of $\phi$ and $\boldsymbol{\theta}$ in the case $H \geq d$ and the proof of Corollary 3 by Zhou (2019), there exists a $\boldsymbol{\theta}^{\text{Alg}}$ such that $\mathfrak{R}^T \geq \Omega(\sqrt{dHT})$. The proof is concluded by combining the results of the two cases. $\qquad\square$