# OpenReview forum: "Towards Provably Efficient Learning of Extensive-Form Games with Imperfect Information and Linear Function Approximation"
_ICLR.cc/2024/Conference — Submitted to ICLR 2024_

### Official Review · Reviewer_zdt3 · 2023-10-24

**Soundness:** 3 good
**Presentation:** 2 fair
**Contribution:** 2 fair
**Rating:** 6
**Confidence:** 3

**Summary:**

This paper studies imperfect information EFGs in the setting of partially observable Markov games. Due to the partial observability of the game, current techniques rely on either online learning with loss estimations or Monte Carlo counterfactual regret minimization to achieve convergence. However, the sample complexities depend on the cardinalities of the information set space and action space, which could be extremely large. This paper introduces efficient algorithms in the setting of linear function approximation, which circumvents the problem of dimensionality. In their setting, transitions between states are known and the current information set is also known to the players, and finally the reward functions admit a linear structure. With all this in place, the authors present two algorithms based on mirror descent and FTRL, which the authors call Least Squares OMD and Least Squares FTRL. Both these algorithms admit comparable regret guarantees to the existing state of the art algorithms, albeit with modified dependencies on exploration parameters and game tree structure respectively.

**Strengths:**

Overall, the paper has a clear structure and the extension of prior work to the linear function approximation setting is a reasonable and interesting setting. Despite the complexity of the arguments required and the heavy notation, the technical work in the paper is impressive, and the authors do an admirable job with making the paper readable. The motivation for avoiding the curse of dimensionality in POMGs is also strong, and this paper is indeed an initial foray into understanding learning in IIEFGs with linear function approximation.

**Weaknesses:**

The main weakness of the paper is that while the overarching setting is clear (POMG, linear function approximation etc), there are many other instances where the writing is less clear. For instance, the introduction describes the regret bounds of previous work without defining X or A. The definition of parameters $\alpha$ and $\lambda$ in the regret bounds of LSOMD and LSFTRL are similarly obfuscatory and not well explained - how the bound compares to prior work of Bai et al and Feigel et al is merely alluded to but not properly substantiated with experiments or examples. Along this line, Assumptions 3.2 and 4.1 are not clearly explained to the reader - how restrictive are these assumptions in practice? I feel that some experiments on games with large action spaces might have been helpful to express the relative advantages of using the approach in this paper. Overall, I believe if some explanatory remarks were added that could clear up some of the confusion, and a writing pass was done to make some of the explanations and contributions clearer, then the paper would be more worthy of acceptance.

**Questions:**

- How restrictive are Assumptions 3.2 and 4.1 in practice?
- Is there a multiplayer variant of the algorithms proposed that can provide convergence guarantees in multiplayer, large-scale games? Does the need for each player to have access to the current information set make such a claim invalid?

---

> ### Author Response · Authors · 2023-11-20
> **Author Response 1**
>
> Thank you for your valuable comments and suggestions. We provide our response to each question in turn below.
>
> **Q1. "For instance, the introduction describes the regret bounds of previous work without defining X or A."**
>
> We apologize for the incomplete descriptions. We have now revised our paper to add more descriptions about the corresponding notions for better readability. Please see Section 1 in the revised version of our paper (highlighted in green).
>
> **Q2. "The definition of parameters $\alpha$ and $\lambda$ in the regret bounds of LSOMD and LSFTRL are similarly obfuscatory and not well explained."**
>
> We have now added more intuitions and explanations about the parameters $\rho$ (here $\rho$ is a more general and more adaptive parameter than its previous counterpart $\alpha$; please see Assumption 3.2 in the revision of our paper for details) and $\lambda$ for better readability. In particular, we have additionally showcased two examples to demonstrate different possible values of $\lambda$. Please see Section 3.3 and 4.2 and Appendix E.1 in the revision of our paper (highlighted in green).
>
> **Q3. "how the bound compares to prior work of Bai et al and Feigel et al is merely alluded to but not properly substantiated with experiments or examples."**
>
> We have now added more discussions and comparisons between our results and previous results for learning tabular partially observable Markov games (POMGs). Please see Remark 3.4, 4.3, and 4.5 in the revision of our paper (highlighted in green). In addition, we now provide a regret lower bound of order $\Omega(\sqrt{d\min(d,H)T})$. Please see Section 4.3 for the formal statements and the discussions on the tightness of our regret upper bounds in the revised version of our paper (highlighted in green).
>
> **Q4. "Along this line, Assumptions 3.2 and 4.1 are not clearly explained to the reader - how restrictive are these assumptions in practice? I feel that some experiments on games with large action spaces might have been helpful to express the relative advantages of using the approach in this paper."**
>
> We believe that Assumption 3.2 is a relatively moderate assumption, which only requires the minimum eigenvalue of the feature covariance matrix $\boldsymbol{Q}_{\pi^t, h}$ generated by the exploration policy $\pi_t$ is lower bounded and might not be hard to be satisfied in practice. Also, nearly the same assumption (more specifically, Assumption 2 in [1] and Assumption 3 in [2]) has been adopted in the literature of RL with linear function approximation. For Assumption  4.1, $\lambda$ will not be very large if the environment transition and the opponent's policy is relatively "balanced" and can only be as bad as $\lambda=\mathcal{O}(X)$ and this case can only happen in very extreme circumstances (please see Section 4.2 and Appendix E.1 in our revised paper for details). Therefore, we think these two assumptions are generally not very restrictive in practice.
>
> Besides, we do think conducting empirical evaluations to further validate the performance of our proposed algorithms is valuable and is an important next step. However, we also would like to note that in this work we aim to make the first step to advance the theoretical understandings of learning partially observable Markov games (POMGs) in the function approximation setting. Thus we did not include empirical evaluations in our paper following previous works [3,4] and we leave conducting empirical evaluations of our proposed algorithms as one of our future studies.
>
> **Q5. "Overall, I believe if some explanatory remarks were added that could clear up some of the confusion, ... then the paper would be more worthy of acceptance."**
>
> Thank you again for pointing this out. As mentioned above, we have now added more descriptions of the related notions in Section 1, more intuitions and discussions about the two assumptions, and more comparisons between our results and previous ones in Section 3.3, 4.2, and Appendix E.1, and the discussions between our regret upper bounds and lower bound in Section 4.3, in the revision of our paper. We hope this can address your concern regarding the presentation of our paper.

---

> > ### Comment · Reviewer_zdt3 · 2023-11-20
> > **Response to Author Comments**
> >
> > Thank you for the changes and explanation to my concerns. In particular the change to Assumption 3.2 increases my belief about the practical utility of LSOMD. I think the paper is certainly in a better state and clearer in its exposition, however I still do not have a strong leaning towards accepting the paper due to the lack of empirical evidence corroborating the algorithms presented. As such I have increased the score marginally.

---

> > > ### Author Response · Authors · 2023-11-20
> > > **Author Response 3**
> > >
> > > Thank you for your positive feedback and raising your score for our work! We are currently working on conducting empirical evaluations of our proposed algorithms. We will be sure to incorporate the empirical evaluations into the future version of our work.

---

> ### Author Response · Authors · 2023-11-20
> **Author Response 2**
>
> **Q6. "Is there a multiplayer variant of the algorithms proposed that can provide convergence guarantees in multiplayer, large-scale games? Does the need for each player to have access to the current information set make such a claim invalid?"**
>
> Thank you for referring to this! In multiple-player general-sum POMGs, the linear structure might need to be reformulated by considering some kind of linear completeness of the local $Q$-value function for each player, to avoid the exponential dependence on the number of players (*i.e.*, "the curse of multiagents"), as multiple-player general-sum (fully observable) Markov games [5,6]. We believe that it is feasible (and also interesting) to extend our proposed algorithms and analyses from the setting of two-player zero-sum POMGs to the setting of multiple-player general-sum POMGs, which we leave as our future study. Besides, in multiple-player general-sum POMGs, we think each player has access to their own current infoset is still a reasonable problem setup and does not make the above claim invalid. For instance, even in Poker games involving more than two players, each player always has access to their own cards in hand, which are exactly their infosets in the game considered.
>
> [1] Neu et al. Online learning in mdps with linear function approximation and bandit feedback. NeurIPS, 2021.
>
> [2] Luo et al. Policy optimization in adversarial mdps: Improved exploration via dilated bonuse. NeurIPS, 2021.
>
> [3] Kozuno et al. Model-Free Learning for Two-Player Zero-Sum Partially Observable Markov Games with Perfect Recall. NeurIPS, 2021.
>
> [4] Bai et al. Near-Optimal Learning of Extensive-Form Games with Imperfect Information. ICML, 2022.
>
> [5] Wang et al. Breaking the Curse of Multiagency: Provably Efficient Decentralized Multi-Agent RL with Function Approximation. COLT, 2023.
>
> [6] Cui et al. Breaking the Curse of Multiagents in a Large State Space: RL in Markov Games with Independent Linear Function Approximation. COLT, 2023.

---

### Official Review · Reviewer_AopK · 2023-10-25

**Soundness:** 3 good
**Presentation:** 3 good
**Contribution:** 3 good
**Rating:** 6
**Confidence:** 2

**Summary:**

The paper introduces two algorithms for online learning in Partially Observable Markov Games: Linear Stochastic Online Mirror Descent (LSOMD) and Least-Squares Follow-The-Regularized-Leader (LSFTRL). This research uniquely emphasizes learning in Imperfect Information Extensive-Form Games using linear function approximation, diverging from traditional settings. A significant contribution is the novel least-squares loss estimator that leverages composite reward features. For LSOMD, the research employs an exploration policy to derive its regret bound, denoted as $\tilde{O}(\sqrt{HX^2d\alpha^{-1}T})$, contingent on a specific quantity $\alpha$ related to the exploration policy. Furthermore, LSFTRL adopts a "balanced transition" methodology, previously used in several works, for its loss estimator. This results in regret bounds $\tilde{O}(\sqrt{H^2d\lambda T})$ and $\tilde{O}(\sqrt{HXd T})$ ($\lambda\geq X/H$), which rely on another quantity $\lambda$ linked to the game tree structure.

**Strengths:**

- The paper introduces two novel algorithms for online learning in POMGs, marking the first study of Imperfect Information Extensive-Form Games using linear function approximation. This linear approximation approach is notably more practical.
- The paper clearly presents the problem setting, making it accessible to readers unfamiliar with the topic.
- The algorithms are clearly explained, and the intuition behind them is also provided.

**Weaknesses:**

- For LSOMD, the parameter setting ($\eta$) necessitates prior knowledge of $\alpha$. A discussion on determining or estimating $\alpha$ is required. An adaptive algorithm would be a preferable solution.
- Similarly, the parameter setting ($\eta$) of LSFTRL requires prior knowledge of $\lambda$. Besides lack of discussion on $\lambda$, it is not clear whether we should use the first or the second parameter initialization in practice. I think when $\lambda\geq X/H$, the second initialization should be adopted, but how to determine whether $\lambda\geq X/H$ and what regret we can get if $\lambda< X/H$?
- Discussion on lower bounds is needed, even if some necessary conditions on some of the factors in the regret bounds are helpful.
- No experiment is provided in the main paper.

**Questions:**

- Is it possible to provide an adaptive algorithm that achieves a similar regret bound without knowing $\alpha$ or $\lambda$ in advance? Please refer to the weaknesses part.
- Are the regret bounds provided by LSOMD and LSFTRL optimal? I guess that $\sqrt{HdT}$ is unavoidable in the bound, but this requires a more rigorous proof.
- Could you conduct empirical experiments to demonstrate the algorithms' effectiveness? Even numerical tests using toy examples would be insightful.
- In Assumption 2.1, the paper assumes $\left\|{\bf\theta}\right\|_2\leq \sqrt{d}$ and $\sup_{s_h,a_h,b_h}\left\|\Phi(s_h,a_h,b_h)\right\|_2\leq 1$. Can this pair of inequalities be substituted with $\bar{r}_h(s_h,a_h,b_h)\leq \sqrt{d}$?

---

> ### Author Response · Authors · 2023-11-20
> **Author Response 1**
>
> Thank you for your valuable comments and suggestions. We provide our response to each question in turn below.
>
> **Q1&2&5. Adaptivity of LSOMD and LSFTRL algorithms.**
>
> The dependence on $\rho$ (here $\rho$ is a more general and more adaptive parameter than its previous counterpart $\alpha$; please see Assumption 3.2 in the revision of our paper for details) of the learning rate $\eta$ and exploration parameter $\gamma$ in our LSOMD algorithm comes from the fact that it is required to control the variance of the loss estimate so that the loss estimate is not prohibitively negative when using the negative entropy as the potential function of online mirror descent (OMD). We conjecture that the dependence on $\rho$ of $\eta$ and $\gamma$ (and also the regret guarantee of LSOMD) can be eliminated by further considering leveraging the log-barrier regularizer to tackle arbitrarily negative loss functions, as it is done in (single-agent) reinforcement learning [1,2,3] (and references therein). We leave this extension as our future study.
>
> For the dependence on $\lambda$ of the learning rate $\eta$ of our LSFTRL algorithm, we now provide an alternative initialization of the learning rate $\eta$ in Remark 4.3 of our revised paper, which has no dependence on $\lambda$ at a slight cost of making the regret upper bound in Theorem 4.2 have an additional $\widetilde{\mathcal{O}}(\sqrt{\lambda})$ dependence. Besides, when $\lambda\geq X/H$, as the reviewer mentioned, it suggests that adopting the second initialization of our LSFTRL algorithm is preferable. But it is indeed generally hard to determine in advance whether $\lambda\geq X/H$ or not since $\lambda$ depends on the "reaching probability" of infosets contributed by not only the tree structure and the environment state transitions of the game but also the opponent's policy $\nu_t$. Nonetheless, we believe that this issue might be addressed by utilizing a model section approach [4,5] that runs a master algorithm to control two LSFTRL algorithms with different initialization of learning rate $\eta$ as base algorithms. In this manner, the master algorithm is expected to perform nearly as well as the best base algorithm if the best base algorithm were to be run separately. We leave this extension as our future study.
>
> **Q3&6. "Discussion on lower bounds is needed, even if some necessary conditions on some of the factors in the regret bounds are helpful."**
>
> We now provide a regret lower bound of order $\Omega(\sqrt{d\min(d,H)T})$. Please see Section 4.3 for the formal statements and the discussions on the tightness of our regret upper bounds in the revised version of our paper (highlighted in green).
>
> **Q4&7. No experiment is provided in the main paper.**
>
> We do think conducting empirical evaluations to further validate the performance of our proposed algorithms is valuable and is an important next step. However, we also would like to note that in this work we aim to make the first step to advance the theoretical understandings of learning partially observable Markov games (POMGs) in the function approximation setting. Thus we did not include empirical evaluations in our paper following previous works [6,7] and we leave conducting empirical evaluations of our proposed algorithms as one of our future studies.
>
> **Q8. "In Assumption 2.1, the paper assumes ... Can this pair of inequalities be substituted with $\bar{r}_h(s_h,a_h,b_h)\leq \sqrt{d}$."**
>
> In general, the regularity conditions imposed over $\mathbf{\phi}(\cdot,\cdot,\cdot)$ and $\mathbf{\theta}_h$ can not be substituted with the regularity condition imposed over $|\bar{r}_h(s_h,a_h,b_h)|$, as they are required to control the variance of the loss estimate (please see this in the proof of Lemma D.4 in Appendix D.2.2 of our paper).
>
> [1] Jin et al. Simultaneously Learning Stochastic and Adversarial Episodic MDPs with Known Transition. NeurIPS, 2020.
>
> [2] Jin et al. The best of both worlds: stochastic and adversarial episodic MDPs with unknown transition. NeurIPS, 2021.
>
> [3] Dai et al. Refined Regret for Adversarial MDPs with Linear Function Approximation. ICML, 2023.
>
> [4] Agarwal et al. Corralling a Band of Bandit Algorithms. COLT, 2017.
>
> [5] Dann et al. A Blackbox Approach to Best of Both Worlds in Bandits and Beyond. COLT, 2023.
>
> [6] Kozuno et al. Model-Free Learning for Two-Player Zero-Sum Partially Observable Markov Games with Perfect Recall. NeurIPS, 2021.
>
> [7] Bai et al. Near-Optimal Learning of Extensive-Form Games with Imperfect Information. ICML, 2022.

---

> ### Author Response · Authors · 2023-11-23
> **Author Response 2**
>
> Dear reviewer AopK,
>
> We have now included more discussions on the adaptivity of our LSOMD and LSFTRL algorithms, especially showing that our LSFTRL algorithm can work without the knowledge of parameter $\lambda$. In addition, we now substitute the previous Assumption 3.2 with a more general and more adaptive one, only requiring the feature space of infoset-actions is well explored under the uniform policy, and we provide more intuitions and discussions about the two assumptions in Section 3.3, 4.2 and Appendix E.1. We further provide a regret lower bound of order $\Omega(\sqrt{d \min (d, H) T})$ and also include the discussions about the tightness of our regret upper bounds and more comparisons between our results and previous ones in Section 3.3, 4.2 and 4.3. In this work, we mainly focus on the theoretical understandings of learning IIEFGs/POMGs with linear function approximation, and thus we did not include the empirical evaluations in the previous version of our paper, as most of the previous works studying tabular IIEFGs/POMGs, (fully observable) Markov games with linear function approximation, and other RL problems with linear structures (*e.g.*, linear MDPs, linear mixture MDPs). We are currently doing our best to conduct empirical evaluations of our proposed algorithms, which will take some time, and we will be sure to incorporate the empirical evaluations into the future version of our work.
>
> We thank you again for your tremendously valuable review of our paper. As the author-reviewer discussion period is coming to an end, please let us know if you have any questions about our responses or any further concerns about our paper. If not, we would appreciate it very much if you could consider improving your evaluation of our paper.
>
> Best,
>
> Authors

---

### Official Review · Reviewer_YLD6 · 2023-10-31

**Soundness:** 3 good
**Presentation:** 3 good
**Contribution:** 3 good
**Rating:** 5
**Confidence:** 4

**Summary:**

This paper presents a solution to the problem of learning in two-player zero-sum imperfect information extensive-form games (IIEFGs) with linear functional approximation. The focus is on IIEFGs in the formulation of partially observable Markov games (POMGs) with known transitions and unknown rewards while admitting a linear structure over the reward functions. The challenge is that both players are unaware of the current underlying state, since only the current information set rather than the state is observable. This poses substantial difficulties in exploiting the linear structure of the reward functions, as the current feature corresponding to the current state is unknown.

To address this problem, the paper proposes a linear loss estimator based on the composite features of information set-action pairs. These composite reward features can be seen as features of corresponding information set-actions, and are weighted by the transitions and opponent's policy. The paper proves the unbiasedness of this estimator and derives regret bounds that depend on various game parameters.

**Strengths:**

Disclaimer) I personally research on Markov Game, not on EFG. But I carefully read this paper, checked every proof.

As I try to find the literature of EFG regarding this, there are no linear approximation papers, and it is actually needed. In this sense, this is a good starting point I believe.

**Weaknesses:**

I have several concerns with assumptions. I will write it down in the questions section. My main concern is the tightness of this analysis, too strong assumptions, and also computational issues. If these are resolved, I am willing to change my score. I think this topic is extremely important for EFG literature, while this paper is somewhat weak because of the following questions.

**Questions:**

Q1) I think the assumption that is on page 7 (regarding exploration) is not used in Luo et al and Neu et al. As far as I understand, Assumption 3.2 provides a very strong assumption, basically saying that every policy covers any kind of x. Which is actually making no need for exploration. I think this is very strong, and it is not done in other "recent" works (a similar assumption was at very "traditional" papers) Can you clarify that? I think this is a kind of uniform-coverage assumption in offline RL literature. Or can you refer any specific examples that are used in EFG literature?

Q2) Just want to understand: is this paper making an assumption that we know p(s|a.b)? or do we need to learn that?

Q3) theorem 3.3: Still, that depends on X^2, so in the finite state action space case, it is not optimal. expected regret is not enough. Maybe for the general space, does this algorithm match with the lower bound? or can we prove (or have some clue) the lower bound? Also, can we eliminate H term as Balanced FTRL?

Q4) page 8 : Still computation depends on A.. (O(XA)) which means that it does not scale with the linear representation. This is also related to Q1, as we want to cover the large action space or (maybe infinite size A). That means that alpha is at least smaller than  1/A.

Q5) I am not an author of
Breaking the curse of multiagency: Provably efficient decentralized multi-agent rl with function approximation
and
Breaking the curse of multiagents in a large state space: RL in Markov games with independent linear function approximation
but they are providing linear approximation scheme for Multi-agent RL. What is the relationship between this paper and these two approximations?

**Nov 25) Still I do not think that the assumption is comparable, as this paper assumes a condition about "exploration policy"'s eigenvalue, so I want to re-evaluate.**

---

> ### Author Response · Authors · 2023-11-20
> **Author Response 1**
>
> Thank you for your valuable comments and suggestions. We provide our response to each question in turn below.
>
> **Q1. "I think the assumption that is on page 7 ..."**
>
> Thanks for pointing this out. We fully agree that the previous Assumption 3.2 might be restrictive in some practical scenarios. We have now substituted this assumption with a more general and more adaptive one, which only requires that each direction of the feature space of infoset-actions is well explored under some explorative policy. Also, now the new assumption is more consistent with its counterparts in the previous works studying (single-agent) linear MDPs that we referred to (more specifically, Assumption 2 in [1] and Assumption 3 in [2]). The regret upper bound of our LSOMD algorithm and its proof have been altered to adapt to this new assumption. Please see Section 1, 3.2, 3.3, and Appendix D.2.2 for the revisions in our paper (highlighted in green).
>
> **Q2. "is this paper making an assumption that we know p(s|a.b)? or do we need to learn that?"**
>
> We would like to note that both our algorithm LSOMD and LSFTRL do not need to know the underlying transition $p_h(\cdot\mid s_h,a_h,b_h)$ in the "offline setting", in which the max player has access to the *composite feature vectors* weighted by the unknown transition and opponent's policy. We have now further improved corresponding statements to make this point clear in the revision of our paper (highlighted in green).
>
> **Q3. "theorem 3.3: Still, that depends on X^2, so in the finite state action space case, it is not optimal. ... Also, can we eliminate H term as Balanced FTRL?"**
>
> Indeed, the $\widetilde{\mathcal{O}}(X)$ dependence on the infoset space size is not optimal and the expected regret of our algorithms is currently not sufficient to find the Nash equilibrium (NE) using the conventional regret to NE conversion as mentioned in Section 5. However, we believe that both the algorithmic designs and results of our LSOMD algorithm still have their own merits, especially considering that it is first result for learning partially observable Markov games (POMGs) with linear function approximation, and the analysis of such result is not a straightforward extension of the tabular case and requires to overcome non-trivial technical difficulties when bounding the log-partition function $\log Z_1^t\left(x_1\right)$ as described in Section 3.3. On the other hand, we would like to remark that the result of our second algorithm LSFTRL is $\widetilde{\mathcal{O}}(\sqrt{H^2d\lambda T})$, with no dependence on $X$. Also, we believe that obtaining the high probability version of our result is also possible by further considering the techniques for studying the high probability guarantees of adversarial linear bandits  (*e.g.*, using self-concordant barrier potential functions in [3]) as mentioned in Remark 3.4. We leave this extension as our future work.
>
> Besides, we now provide a regret lower bound of order $\Omega(\sqrt{d\min(d,H)T})$, which shows that the dependence on $H$ of regret upper bound generally can not be eliminated in our case, different from the dependence on $H$ in the tabular case. Please see Section 4.3 for the formal statements and the discussions on the tightness of our regret upper bounds in the revised version of our paper (highlighted in green).
>
> **Q4. "page 8 : Still computation depends on A.. (O(XA)) ...."**
>
> We fully agree that in the presence of a particularly large action space the computation of the first initialization of our LSFTRL algorithm is not computationally efficient enough due to the polynomial dependence on $A$. However, we believe the algorithmic design and the result of the first initialization of our LSFTRL algorithm are still valuable and have their own merits since they are the first such algorithms and results for this challenging problem and the statistical complexity (*i.e.*, regret guarantee) of LSFTRL has no dependence on both $A$ and $X$. In addition, we would also like to note that both the statistical and computational complexities of the second initialization of our LSFTRL algorithm have no polynomial dependence on $A$. For the dependence on $\alpha$ in the previous version of our paper, please see our response to Q1.

---

> ### Author Response · Authors · 2023-11-20
> **Author Response 2**
>
> **Q5. "What is the relationship between this paper and these two approximations?"**
>
> Our work departs from theirs mainly in the following two aspects. They study Markov games with perfect information, *i.e.*, fully observable Markov games. While in the setting of our POMG problem, the underlying state is not observable. Instead, in our problem, the players only have access to the infosets, which are partitions of the state space. On the other hand, they study multi-player general-sum Markov games and we study two-player zero-sum Markov games. Another minor difference is that we only assume the rewards are linearly realizable while they assume some kind of linear completeness of the local $Q$-value function of each player involved in the game.
>
> [1] Neu et al. Online learning in mdps with linear function approximation and bandit feedback. NeurIPS, 2021.
>
> [2] Luo et al. Policy optimization in adversarial mdps: Improved exploration via dilated bonuse. NeurIPS, 2021.
>
> [3] Lee et al. Bias no more: high-probability data-dependent regret bounds for adversarial bandits and mdp. NeurIPS, 2020.

---

> ### Comment · Reviewer_YLD6 · 2023-11-20
> **Response**
>
> Thank you for your thoughtful revisions and the effort put into addressing my previous concerns. I appreciate the depth of your analysis, particularly regarding the lower bound aspect, which presents an intriguing perspective.
>
> However, I still have a concern about the assumptions made on page 7 of your paper. Upon closer examination, it seems that these assumptions do not align well with those in references [1] and [2]. In reference [1], the assumption is specifically tied to a 'uniform' (one) policy, rather than every or exploratory policy. Meanwhile, reference [2] seems to focus more on the aspect of low rank. Given these differences, I'm not convinced that mentioning your paper alongside these works is appropriate.
>
> I believe this might stem from a misunderstanding of the previous literature, rather than an intentional oversight. To ensure the integrity and scientific rigor of this work, authors should thoroughly understand and accurately represent the assumptions and findings of related studies.
>
> Additionally, I have a question regarding the IIEFG case: Could you explain why we can eliminate the dependency on $H$ in this scenario, whereas, in your paper, a lower bound related to $H$ is established? Understanding this distinction could provide valuable insight into the nuances of your approach.
>
> Thank you for your ongoing dialogue on these matters. I look forward to your response and further clarification.

---

> ### Author Response · Authors · 2023-11-20
> **Author Response 3**
>
> Thank you for your quick response! For Assumption 3.2, we believe there might be some misunderstandings in the previous statements of this assumption and there are indeed some nuances between our previous assumption and the ones in single-agent RL literature [1,2], as we now realize. However, we would also like to note that actually, the same regret of our LSOMD algorithm can be established under the assumption that the minimum eigenvalue of the feature covariance matrix $\bf{Q}_{\pi,h}$ generated by the (one) uniform policy $\pi$ is lower bounded. We have now further revised the corresponding statements of this assumption and also altered some descriptions of our LSOMD algorithm to adapt to the new statements of this assumption. Please see Section 3.2 and 3.3 for the revisions.
>
> For the $\Omega(\sqrt{H})$ dependence in our regret lower bound, it suffices to consider the case of $H\leq d$, in which the hardness of learning our problem is essentially the same as learning a stochastic linear bandit problem with $2^H$ distinct feature vectors of arms, leading to a lower bound of order $\Omega(\sqrt{dHT})$. While in tabular case, the hardness of learning IIEFGs/POMGs is the same as learning a multi-armed bandit problem with $A^H=A^{H-1}\cdot A=\mathcal{O}(XA)$ arms [3,4], leading to an $\Omega(\sqrt{XAT})$ regret lower bound. Intuitively, the factor $H$ does not appear in the lower bound of the tabular case does not mean that $H$ imposes no effects on the hardness of learning this game in the tabular case, as $\mathcal{O}(XA)\geq H$ always holds. Moreover, it is also not unusual to see that the dependence on $H$ in sequential decision-making problems with linear structures is larger than it is in the tabular case of the sequential decision-making problems. For instance, the minimax optimal regret lower bound for learning tabular MDPs is $\Omega(H\sqrt{ S A T})$ [5] (note that here we denote by $T$ the number of episodes), while the minimax optimal regret lower bounds for learning linear MDPs and linear mixture MDPs are $\Omega(dH^{3/2}\sqrt{T})$ [6].
>
> We hope our responses can help address your concerns and we are also more than happy to answer any further questions.
>
> [1] Neu et al. Online learning in mdps with linear function approximation and bandit feedback. NeurIPS, 2021.
>
> [2] Luo et al. Policy optimization in adversarial mdps: Improved exploration via dilated bonuse. NeurIPS, 2021.
>
> [3] Bai et al. Near-Optimal Learning of Extensive-Form Games with Imperfect Information. ICML, 2022.
>
> [4] Fiegel et al. Adapting to game trees in zero-sum imperfect information game. ICML, 2023.
>
> [5] Azar et al. Minimax Regret Bounds for Reinforcement Learning. ICML, 2017.
>
> [6] Zhou et al. Nearly Minimax Optimal Reinforcement Learning for Linear Mixture Markov Decision Processes. COLT, 2021.

---

> ### Comment · Reviewer_YLD6 · 2023-11-21
> **Great!!**
>
> I read the paper again, and I realized that this paper substantially improved after its revision. Therefore, I am leaning this paper to be accepted. This version became a complete version compared to the previous manuscript.

---

> > ### Author Response · Authors · 2023-11-21
> > **Author Response 4**
> >
> > Thank you once again for such a valuable review and for raising your score for our work!

---

### Official Review · Reviewer_ahTG · 2023-10-31

**Soundness:** 3 good
**Presentation:** 3 good
**Contribution:** 2 fair
**Rating:** 6
**Confidence:** 3

**Summary:**

The paper studies algorithms for imperfect information extensive form games (IIEFGs) with linear function approximation, formulated as partially observable Markov games (POMGs) with known transition and bandit feedback. Least-squares estimators are proposed and incorporated into online mirror descent (OMD) and follow-the-regularized-leader (FTRL). Regret bounds are provided for both algorithms.

**Strengths:**

The setting studied in the paper is meaningful and interesting. While it seems like a natural generalization that stems from POMDPs and MGs, the results on linear POMGs in this paper are original and nontrivial. To my knowledge, these are the first algorithms and regret bounds for such linear POMGs.

**Weaknesses:**

I have some slight concern over the significance of the results: On one hand, the results (in particular, the regret bounds) at a high level do not seem surprising or particularly insightful (although there may be things I missed); on the other hand, this is clearly a rather generic framework for a POMGs, which in reality are usually highly complex (e.g., $H$ and $X$ can be large and $\alpha$ tiny) but often come with specific structure that can be leveraged. Thus, I am not fully convinced that the algorithms and results in this paper can be useful in real world settings, although this by no means diminishes the theoretical value of the results. In addition, the specific assumptions on the structure (e.g., the linearity of rewards and access to the opponent's feature vectors) seem to make the results less general.

**Questions:**

A minor comment: I would appreciate if the author(s) define the relevant quantities upfront in the introduction (I see them in the abstract and later in the text, but think they deserve a note when first mentioned).

---

> ### Author Response · Authors · 2023-11-20
> **Author Response 1**
>
> Thank you for your valuable comments and suggestions. We provide our response to each question in turn below.
>
> **Q1. "the results (in particular, the regret bounds) at a high level do not seem surprising or particularly insightful"**
>
> We would like to note that our results are not straightforward extensions of their tabular counterparts. In specific, as mentioned in Section 3.3, establishing the regret guarantee of our LSOMD requires to bound the summation of the log-partition function $\log Z_1^t\left(x_1\right)$, which is more difficult than bounding this term in the tabular cases and intrinsically requires new ingredients in the analysis. Moreover, the regret analysis of the LSFTRL algorithm is also not a straightforward extension of the tabular case. Particularly, we use a ``balanced" transition that is not used in previous works and a refined analysis is established to further shave a factor of $\mathcal{O}(\sqrt{A})$, as described in Section 4.2.
>
> **Q2. "this is a rather generic framework for POMGs, which in reality are usually highly complex ... although this by no means diminishes the theoretical value of the results"**
>
> Indeed, in practice, one may consider using more adaptive algorithms, with tailored components to leverage specific structures in specific problem instances. However, we would like to note that as previous works establishing worst-case theoretical guarantees for partially observable Markov games (POMGs) [1,2,3], we also aim to devise the general algorithmic framework based on online mirror descent (OMD) and follow-the-regularized-leader (FTRL). Besides, though there are cases where $H$ and $X$ might be large and $\rho$ is small (here $\rho$ is a more general and more adaptive parameter than its previous counterpart $\alpha$; please see Assumption 3.2 in the revision of our paper for details), we believe that the result of our LSOMD algorithm still has its own merits, especially considering that it leads to the first theoretical guarantee for this challenging problem and establishing such result requires non-trivial ingredients in the analysis as mentioned in Section 3.3. Further optimizing the dependence on factors $H$, $X$ and $\rho$ in the result of our LSOMD algorithm is interesting and also challenging, which we leave as our future works. Moreover, we note that the result of our second algorithm LSFTRL with the first choice of the ``balanced" transition is $\widetilde{\mathcal{O}}(\sqrt{H^2d\lambda T})$, without dependence on $X$.
>
> **Q3. "the specific assumptions on the structure (e.g., the linearity of rewards and access to the opponent's feature vectors) seem to make the results less general"**
>
> We fully agree that assuming the linear realizability over the reward functions might be restrictive in some practical scenarios. However, we believe that studying linear realizability assumption is a reasonable first step for POMGs in the function approximation setting, as it is also the reasonable first step to study linear realizability assumption for other more amenable sequential decision-making problems including bandits and (single-agent) reinforcement learning in the function approximation setting. Moreover, though we study POMGs with linear function approximation, we think the algorithmic designs and results of both our LSOMD and LSFTRL algorithms are still valuable as the first algorithms and results for learning POMGs in the function approximation setting. Also, we believe that our algorithmic designs and results may further serve as a building block for potential extensions into POMGs with general function approximation (*e.g.*, low-rank structures [4] and general complexity measures [5,6,7,8]). We leave extending the analyses of POMGs with linear function approximation in this work to POMGs with general function approximation as our future study.
>
> The assumption that the max-player has access to the feature vectors of state-actions weighted by the opponent's policy (and environment transition) is because no players in POMGs can see the current underlying state $s_h$ of the system, let alone regress reward $r_h\left(s_h, a_h, b_h\right)$ against feature vector $\phi\left(s_h, a_h, b_h\right)$ of the state-actions. Thus it is currently highly unclear how to eliminate this assumption due to the partial observability of this problem. Further eliminating this assumption is interesting and also challenging, and we leave the investigation into whether it is possible to eliminate this assumption as our future study.

---

> ### Author Response · Authors · 2023-11-20
> **Author Response 2**
>
> **Q4. "I would appreciate if the author(s) define the relevant quantities upfront in the introduction"**
>
> Thanks again for pointing this out. We have now further added more descriptions and explanations about $\rho$ (a refined parameter of its previous counterpart $\alpha$), $\lambda$ and other related notions and further polished the writing for better readability. Please see these revisions in the introduction section of our paper (highlighted in green).
>
> [1] Kozuno et al. Model-Free Learning for Two-Player Zero-Sum Partially Observable Markov Games with Perfect Recall. NeurIPS, 2021.
>
> [2] Bai et al. Near-Optimal Learning of Extensive-Form Games with Imperfect Information. ICML, 2022.
>
> [3] Fiegel et al. Adapting to game trees in zero-sum imperfect information game. ICML, 2023.
>
> [4] Uehara et al. Representation Learning for Online and Offline RL in Low-rank MDPs. ICLR, 2022.
>
> [5] Russo et al. Eluder dimension and the sample complexity of optimistic exploration. NeurIPS, 2013.
>
> [6] Jiang et al. Contextual Decision Processes with low Bellman rank are PAC-Learnable. ICML, 2017.
>
> [7] Jin et al. Bellman Eluder Dimension: New Rich Classes of RL Problems, and Sample-Efficient Algorithms. NeurIPS, 2021.
>
> [8] Du et al. Bilinear Classes: A Structural Framework for Provable Generalization in RL. ICML, 2021.

---

> ### Author Response · Authors · 2023-11-23
> **Author Response 3**
>
> Dear reviewer ahTG,
>
> We have now revised our paper to incorporate more descriptions and discussions about the relevant quantities in Section 1. In addition, we now substitute the previous Assumption 3.2 with a more general and more adaptive one, only requiring the feature space of infoset-actions is well explored under the uniform policy, and we provide more intuitions and discussions about the two assumptions in Section 3.3, 4.2 and Appendix E.1. We further provide a regret lower bound of order $\Omega(\sqrt{d \min (d, H) T})$ and also include the discussions about the tightness of our regret upper bounds and more comparisons between our results and previous ones in Section 3.3, 4.2 and 4.3.
>
> We thank you again for your tremendously valuable review of our paper. As the author-reviewer discussion period is coming to an end, please let us know if you have any questions about our responses or any further concerns about our paper. If not, we would appreciate it very much if you could consider improving your evaluation of our paper.
>
> Best,
>
> Authors

---

### Meta-Review · Area_Chair_9xKQ · 2023-12-06

**Metareview:**

This was a borderline paper. While one of the reviewers mentioned that the authors' responses clarified their questions regarding soundness, they decreased their score again after the end of the rebuttal period after finding another point problematic. The authors were able to respond privately through the AC/author period, and their remarks look convincing to me. I privately reached out to the reviewer to confirm, and the response seems to have resolved most of the reviewer's questions. However, the reviewer still mentioned feeling uncomfortable with the paper.

This paper escalated to the SAC, who kindly helped take a more in-depth look at the paper. Overall, we agreed that it is best not to rush this paper through. In particular, while it is clear that this is a paper with great potential, more than one reviewer feels uneasy recommending acceptance without reservation, and one reviewer felt rather strongly that the current version could be made more complete by including additional results (in particular, experimental validation). Furthermore, we believe that the late-stage back and forth about the remaining technical details was rather time-constrained, and due diligence might require more time before all reviewers feel confident about their assessments.

**Justification For Why Not Higher Score:**

None of the reviewers championed the paper, and some uneasiness related to this paper still persists.

**Justification For Why Not Lower Score:**

N/A

---

### Decision · Program_Chairs · 2024-01-16

Reject